# OpenRCA: Can Large Language Models Locate the Root Cause of Software Failures?

**Junjielong Xu**[1,2*]   **Qinan Zhang**[1]   **Zhiqing Zhong**[1]   **Shilin He**[2†]  **Chaoyun Zhang**[2]
**Qingwei Lin**[2]        **Dan Pei**[3]        **Pinjia He**[1†]        **Dongmei Zhang**[2]   **Qi Zhang**[2]
[1]School of Data Science, The Chinese University of Hong Kong, Shenzhen
[2]Microsoft         [3]Tsinghua University
{junjielongxu,qinanzhang,zhiqingzhong}@link.cuhk.edu.cn,
{shilin.he,chaoyun.zhang,qlin,dongmeiz,qizhang}@microsoft.com
peidan@tsinghua.edu.cn
hepinjia@cuhk.edu.cn

## Abstract

Large language models (LLMs) are driving substantial advancements in software engineering, with successful applications like Copilot and Cursor transforming real-world development practices. However, current research predominantly focuses on the early stages of development, such as code generation, while overlooking the post-development phases that are crucial to user experience. To explore the potential of LLMs in this direction, we propose OpenRCA, a benchmark dataset and evaluation framework for assessing LLMs' ability to identify the root cause of software failures. OpenRCA includes 335 failures from three enterprise software systems, along with over 68 GB of telemetry data (logs, metrics, and traces). Given a failure case and its associated telemetry, the LLM is tasked to identify the root cause that triggered the failure, requiring comprehension of software dependencies and reasoning over heterogeneous, long-context telemetry data. Our results show substantial room for improvement, as current models can only handle the simplest cases. Even with the specially designed RCA-agent, the best-performing model, Claude 3.5, solved only 11.34% failure cases. Our work paves the way for future research in this direction.

## 1 Introduction

Large language models (LLMs) have recently driven significant advancement of software engineering, with numerous research works and real-world applications impacting both the methodology and practice in software development, such as MetaGPT (Hong et al., 2024), SWE-agent (Yang et al., 2024), OpenDevin (Wang et al., 2024b), Copilot, and Cursor. However, existing efforts focus mostly on the early stages of Software Development Life Cycle (SDLC) while ignoring the *post-development* phases. In practice, maintaining software services and debugging issues during online operations are labor-intensive and error-prone tasks that often require 24/7 on-call support. Online incidents can cost service providers billions of dollars (CrowdStrike; UniSuper), highlighting the urgent need for more effective solutions in root cause analysis (RCA) to mitigate software issues.

In recent years, AI researchers have explored various learning-based methods for RCA with techniques such as causal discovery (Arnold et al., 2007; Li et al., 2022a; Chakraborty et al., 2023; Bi et al., 2024), dependency graph analysis (Wang et al., 2023a; Zheng et al., 2024), and other neural networks Wang et al. (2023b); Yu et al. (2021). However, RCA remains challenging due to the immense complexity of real-world software systems, which require multi-step reasoning capabilities over vast and heterogeneous data to identify root causes across diverse failure patterns. As the success of LLMs in software development (Jimenez et al., 2024; Ni et al., 2024; Chen et al., 2023; 2024a) , an important question is: *Can current LLMs be effective in solving RCA challenges?* The answer is critical to further enhance the automation of the entire software lifecycle via LLMs.

---

*Work was done when Junjielong Xu was interning at Microsoft DKI.
†Shilin He and Pinjia He are the corresponding authors.

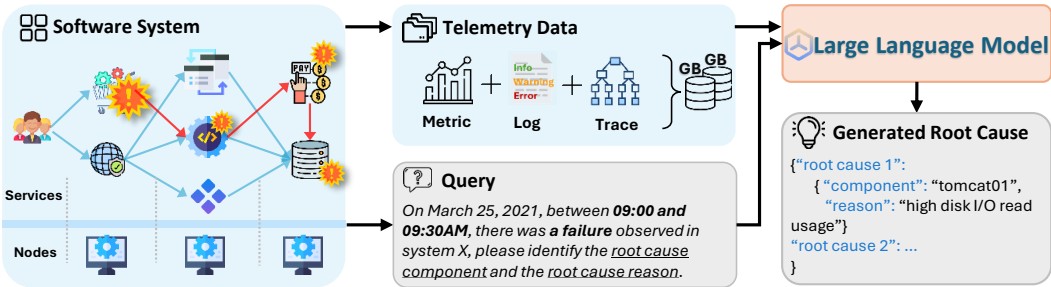

Figure 1: Failures often propagate between services, requiring extensive telemetry (metrics, logs, traces) to identify the root cause. OpenRCA collects real-world failures and corresponding telemetry, framing root cause analysis as a goal-driven task: the model must identify the root cause elements (time, component, reason) specified in the query.

To answer this question, we propose OpenRCA, a public benchmark dataset and evaluation framework for assessing LLMs' root cause analysis ability in a practical software operating scenario. OpenRCA consists of 335 failure cases collected from three heterogeneous software systems deployed in the real world, accompanied by over 68 GB of de-identified telemetry data. Specifically, as shown in Figure 1, each failure case is paired with a query in natural language, requiring LLMs to analyze massive telemetry data to generate the corresponding root cause elements, including time, component, and reason. The process demands LLMs understand intricate system dependencies and conduct complex reasoning across telemetry data of diverse types, such as time series, dependency graphs, and semi-structured text.

Evaluating state-of-the-art LLMs on OpenRCA reveals significant challenges: these models are currently only capable of solving parts of the simplest tasks. For example, Claude 3.5 only resolved 5.37% OpenRCA tasks when oracle telemetry was given. This results further drops to 3.88% when using balanced sampling strategy to extract the possibly related telemetry.

To outline a possible direction for solving OpenRCA tasks, we further developed RCA-agent, a multi-agent system Qiao et al. (2023); Zhang et al. (2024a) based on program synthesis & execution. By utilizing Python for data retrieval and analysis, the model is freed from processing large telemetry as an overly long context. This allows the model to focus solely on reasoning and makes it scalable for massive telemetry. With RCA-agent, the accuracy of Claude 3.5 is further improved to 11.34%.

We believe OpenRCA will serve as a foundational benchmark, driving future research at the intersection of AI and Software Engineering, and allowing the community to explore the true potential of LLMs in solving real-world service reliability problems.[1]

## 2 OPENRCA

### 2.1 PRELIMINARIES OF ROOT CAUSE ANALYSIS

**Root Cause Analysis (RCA)**: In the software development lifecycle, *root cause analysis* refers to the process of identifying the underlying causes of failures, such as service unavailability, in a software system. On-call engineers must gather relevant *telemetry data* and other pertinent information to understand how the failure occurred.

Typically, a root cause should consist of the *originating component* (i.e., which part of the system failed), the *start time* (i.e., when the failure occurred), and the *failure reason* (i.e., why it failed, such as CPU overload or excessive disk throughput). Furthermore, a failure in the originating component can propagate to other components through service dependencies (e.g., a payment service relying on a database) or deployment configurations (e.g., containers on the same server). This propagation can lead to broader system failures, complicating the identification of the exact root cause.

---

[1]The OpenRCA code and data are available at GitHub.

**Telemetry:** Telemetry refers to the data used to monitor the internal status of software systems, encompassing metrics, traces, and logs. Metrics represent time series data points that track key performance indicators (KPIs), such as CPU usage or response time. Traces capture the interactions among multiple system components, illustrating their dependencies, and often structured as a graph. Logs record runtime events and messages for each component, with verbosity levels such as info, warn, and error. Examples of telemetry data are provided in Appendix A.3.

## 2.2 FEATURES OF OPENRCA

OpenRCA is a benchmark designed to evaluate the capability of LLMs to perform RCA in practical software operation scenarios. The benchmark comprises 335 *failure cases* along with associated *telemetry data*, collected from three real-world software systems. Each failure case is structured as a goal-driven RCA task, where a natural language query serves as the input, and the objective is to identify the root causes of the failure. OpenRCA offers the following unique features:

**Real-world Software Development Scenarios:** RCA is a critical step in the software development lifecycle. Current RCA datasets Li et al. (2022a); Ikram et al. (2022) are either synthetic or small-scale. OpenRCA addresses this gap by providing hundreds of failures collected from three real-world software systems. This paves the way for solving more practical RCA problems at scale.

**Goal-driven Task Design:** Traditional RCA datasets Li et al. (2022c); Lee et al. (2023); Yu et al. (2023) often focus on a single goal (e.g., identify the originating component only), resulting in RCA methods tailored to each dataset with low generalizability. OpenRCA adopts a goal-driven approach to cover various aspects of RCA by synthesizing queries in natural language, making RCA a unified task and more accessible for language models. In addition, OpenRCA includes numerous real-world failures, addressing the limitations of traditional synthetic or small datasets used for specific tasks.

**Extensive and Heterogeneous Data:** The failure cases in OpenRCA encompass diverse patterns, such as CPU/memory/network issues across container/node/service levels (see Table 5 in Appendix). Each case involves vast and heterogeneous telemetry data: metrics are time series of numerical values, traces use a graph structure to show dependencies, and logs are semi-structured text, requiring LLMs to make reasoning across diverse data formats.

**Comprehensive LLM Assessment:** OpenRCA requires LLMs to understand the software architecture, interpret various types of real-world telemetry data, and correlate clues and observations from different data pieces. This process assesses LLMs' ability in understanding, reasoning, and decision-making, extending beyond the scope of many existing software engineering tasks.

**Benchmark Updatable:** Our framework for constructing the benchmark allows new labels and telemetry data to be easily integrated into OpenRCA as additional datasets. We also plan to keep OpenRCA updated to maintain its challenge and prevent data contamination.

## 2.3 TASK FORMULATION

**Task Input & Output:** As mentioned earlier, a root cause can be depicted with three elements: *originating component*, *start time*, and *failure reason*. In OpenRCA, we formulate seven tasks (or goals) by combining subsets of these three elements as the target output, which are common in RCA scenarios Lin et al. (2018); Amar & Rigby (2019); Li et al. (2022b). Among the seven tasks, three focus on identifying only a single element, three on identifying two elements, and one on identifying all three root cause elements. Detailed input-output specifications are provided in Appendix A.6. As shown in Figure 1, for each failure case in the benchmark, the input consists of a natural language query and the associated telemetry data, while the output can be one of the seven goals, i.e., a subset of the three root cause elements, in a structured `JSON` format.

**Evaluation:** For each failure case in OpenRCA, it receives 1 point if all generated root cause elements match the ground truth ones, and 0 points if any mismatch was identified. The overall accuracy is the average score across all failure cases. To avoid evaluation errors caused by differences in textual expressions from LLM generation and ground truth, all possible failure reasons and originating components are provided in the prompt beforehand. Further details are provided in Appendix A.7.

Table 1: Summary of OpenRCA datasets regarding the data size, number of unique root causes, i.e., component (C. for short) and reason (R. for short), and telemetry data size.

| Dataset | Cases Count | #Unique RC C. | #Unique RC R. | Telemetry Data Size | Telemetry Data Lines |
|---------|-------------|-----|-----|-------|-------|
| Telecom | 51 | 15 | 5 | 17.6G | 154M |
| Bank | 136 | 14 | 8 | 26.4G | 248M |
| Market | 148 | 44 | 15 | 24.5G | 121M |
| Total | 335 | 73 | 28 | 68.5G | 523M |

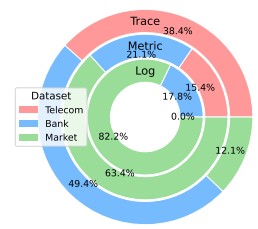

Figure 2: Composition of OpenRCA (by size).

## 2.4 DATASET CONSTRUCTION

OpenRCA consists of three diverse datasets originating from the widely-recognized AIOps Challenge series, held annually since 2018 (AIOps, 2018). Each raw dataset was collected from a real-world software system. However, the raw data is noisy, poorly maintained, and has different input-output specifications, making it unsuitable as a benchmark. Specifically, some datasets lack detailed records of failures like failure reasons, while others are not formulated into input-output pairs required for the RCA task. In addition, disorganized telemetry, inconsistent naming, and failures across systems further reduce data usability. More importantly, there is a significant amount of "dirty" data, such as missing telemetry data, mismatched failure reasons, inconsistent failure times, and incorrect root cause labels. To address these issues, we implemented a four-step data processing procedure with human engagement (Figure 3), as detailed below:

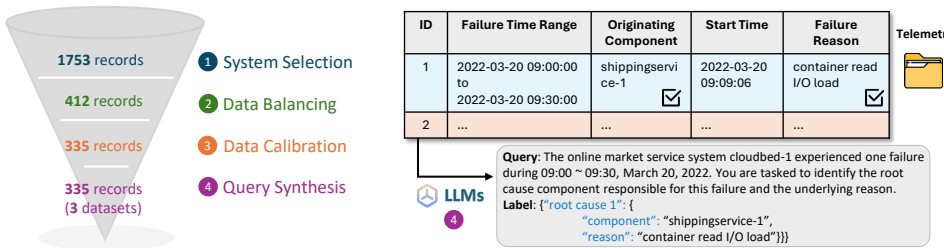

Figure 3: The workflow of OpenRCA benchmark construction.

**Stage 1: System Selection.** Although numerous software systems emerged from past AIOps challenges, many of them either lack labels or only indicate that a failure exists without detailing the root cause. In addition, some systems contain solely metric data, making them not suitable for RCA tasks. After excluding these unqualified systems, we retained data from three systems, including a telecom database system, a banking system, and an online market system, totally comprising 1,753 failure records and their corresponding telemetry. Details of the three systems are in Appendix A.2.

**Stage 2: Data Balancing.** Among the three datasets, the data volumes varied significantly with differences up to 100-fold in scale, which could lead to biases in benchmark evaluation. To alleviate the issue, we downsampled the larger datasets to approximate the scale of smaller ones. Finally, 412 failure records remained, where each dataset contains several tens to a few hundred failure records.

**Stage 3: Data Calibration.** We calibrated the data from two perspectives. First, the raw data followed varying naming conventions as they were collected independently. We standardized the names in all telemetry data and labels, and reorganized the data into a unified format for easier access by models. Second, we employed three experienced engineers to manually verified whether the root cause labels could be pinpointed using the associated telemetry. Failure records were removed if (1) no root cause could be identified from telemetry, (2) telemetry for the failure period was missing, or (3) the root cause inferred from the telemetry misaligned with the labeled one. After filtering, 335 failure records remained with over 68GB telemetry, with each containing about 20GB (distribution shown in Table 1 and Figure 2). More details about data calibration are presented in Appendix A.5.

**Stage 4: Query Synthesis.** Following the goal-driven task design, OpenRCA aims to comprise various goals and their combinations to better generalize to practical scenarios. In this step, we synthesize queries in natural language from failure records because (1) real failure queries are usually unavailable, (2) synthesizing queries ensures diversity and closely resembles human queries, and (3) it is easy to scale up after accommodating new datasets. Specifically, seven goals can be derived from three root cause elements ($C_3^1 + C_3^2 + C_3^3 = 7$), covering various RCA scenarios.

As illustrated in Figure 3, for each failure record, we first randomly select one of the seven goals and combine it with the failure time range (a 30-minute window), and the underlying system as a specification. In some cases, multiple root causes may occur sequentially within the same time range. To address this, the number of actual failures is also included in the specification. Each specification is then fed into LLMs like GPT-4 to generate a natural language query (see Appendix A.6 for prompts), which is further verified by human annotators.

## 3 RCA-AGENT

To resolve an OpenRCA task via LLMs, the first key challenge is how to process large volumes of telemetry data. An intuitive solution is to chunk the data into smaller pieces and feed them into the LLMs sequentially. However, this approach is inefficient, costly and sacrifices the global view. Another method is to sample a subset of data, which is more cost-effective but risks losing critical information. The second challenge is that telemetry data is predominantly non-natural language, consisting of numbers and many rare encoded tokens (e.g., GUIDs, error codes), which LLMs are not well-equipped to handle. An alternative way to process the vast amounts of data is by executing code Qiao et al. (2023); Zhang et al. (2024b), which allows all data operations to be conducted programmatically. This approach eliminates the need to embed raw telemetry data into the LLM context, thereby significantly reducing token consumption while maintaining the global view.

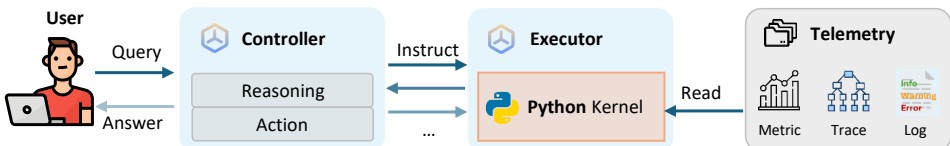

Figure 4: The structure and workflow of RCA-agent.

Based on this idea, we developed *RCA-agent*, a multi-agent system that integrates decision-making, program synthesis, and execution. RCA-agent comprises two LLM agents (prompts are in Appendix B) as described below:

**Controller:** The Controller serves as the main decision-maker, responsible for directing the overall process, analyzing results, and determining next step iteratively. Based on common practices in RCA, it provides high-level guidance for LLMs to follow: *anomaly detection → fault identification → root cause localization*. Additionally, Controller is instructed to analyze in the order of *metric → trace → log*. To help LLMs understand how the telemetry data is organized, the directory structure and telemetry data schema are dynamically injected into the prompt.

**Executor:** The Executor is tasked with writing Python code based on the Controller's instructions, executing the code, and reply back to Controller for further analysis. It comprises two parts: a LLM-based code generator responsible for synthesizing code and a Python kernel environment for executing it. Both the model and the Python environment are stateful, meaning that the generated code and executed variables are cached in memory until the query is resolved.

**Workflow of RCA-agent:** When receiving the input query, Controller follows the general guidance to carry out a series of ReAct reasoning steps. Initially, Controller instructs the Executor to load the relevant telemetry data, and the Executor generates and executes the code. If the execution fails, Executor will reflect itself using the error stack. Otherwise, Executor responds with the results to Controller, who then observes, gains insights, and decides the next action. This iterative process between Controller and Executor continues until all reasoning steps are completed and a final conclusion is reached.

It is important to note that while the RCA-agent uses code generation to avoid processing large amounts of context, allowing for scalability and handling vast amounts of telemetry, its performance is limited by the model's error handling ability. If the model struggles with handling errors, the RCA-agent's effectiveness might be constrained, as will be discussed in Section 5.

# 4 EXPERIMENTAL SETUP

In this section, we describe the experimental setup used to evaluate LLMs on OpenRCA problems.

## 4.1 SAMPLING-BASED METHODS

Given the vast volume of telemetry data, it is impractical to feed all telemetry into the LLMs due to their limited context window. A common strategy to reduce telemetry volume in RCA is sampling Huang et al. (2024); He et al. (2023). Thus, we downsample all telemetry data (including trace, log, and metric) to a frequency of one minute by selecting the first recorded value within each minute, regardless of the original frequency. Meanwhile, we relax the accuracy criteria for predicting failure start time to within one minute of the actual event. However, this sampling is still insufficient as the metrics consist of a large number of KPI types (e.g., memory usage, network delay), necessitating further sampling of KPI types. Thus, we consider two sampling strategies:

**Oracle Sampling:** To investigate the upper bound of the sampling-based method's performance, we introduce the oracle sampling. During benchmark construction, engineers identified a fixed set of "golden" KPIs that are helpful for identifying the root cause. In the oracle sampling, we filtered these "golden" KPIs as our target. Although this approach is unrealistic, it significantly reduces the number of KPIs by 95% (from 1263 to 53), thereby lowering the task complexity. It is important to note that these "golden" KPIs are not the only possible indicators that could point out the root cause. Different KPIs within the same metric file are often correlated; for example, "CPU-used-percent" and "CPU-used-MB" tend to show similar trends. Therefore, even KPIs not included in the fixed set (non-oracle KPIs) may also potentially uncover the failure causes.

**Balanced Sampling:** We use stratified sampling by iteratively selecting one random but unique KPI type from each metric file until the number of sampled KPIs matches that in the Oracle setting. It is a practical and intuitive approach to adapt LLMs to OpenRCA queries, which also ensures balanced representation across all KPI types. Due to correlations between KPIs, balanced random sampling usually produces stable statistical results, with a low variation of 1% in our experiments. To ensure reproducibility, we tested each balanced sampling method three times and reported the median result.

In addition to sampling, we also removed uninformative data columns like service alias, and compressed data that could consume excessive tokens like trace IDs. After sampling, the prompt of these two methods contains around 100K tokens (tokenized by GPT-4o).

## 4.2 LANGUAGE MODELS

To solve OpenRCA tasks, which require processing long contexts, we selected six models with at least 128K token context windows, including three proprietary models: Claude 3.5, GPT-4o, and Gemini 1.5 Pro; and three open-source models: Mistral Large 2, Command R+, and Llama3.1 Instruct. The model checkpoints are shown in Appendix C.1

# 5 EXPERIMENTAL RESULT

This section presents the evaluation results, followed by an analysis and key insights. Table 2 summarizes the accuracy of different methods on the OpenRCA benchmark, where the bold font represents the accuracy of the best model in each column. Overall, proprietary models consistently outperform open-source models across all methods. In addition, RCA-agent performs better than sampling-based approaches, with oracle sampling outperforming balanced sampling. This is expected as oracle sampling leverages the ground-truth information. However, all models face significant challenges in resolving OpenRCA tasks. The best-performing model, Claude 3.5, achieved

Table 2: Accuracy comparison of models using sampling and agent-based methods on OpenRCA (%). `Correct` denotes the percentage of fully solved queries, and `Partial` indicates the percentage where at least one required element is correct.

|  | Model | Balanced | | Oracle | | RCA-agent | |
|---|---|---|---|---|---|---|---|
|  |  | Correct | Partial | Correct | Partial | Correct | Partial |
| Closed | Claude 3.5 Sonnet | 3.88 | 18.81 | 5.37 | 17.61 | **11.34** | 17.31 |
|  | GPT-4o | 3.28 | 14.33 | 6.27 | 15.82 | 8.96 | **17.91** |
|  | Gemini 1.5 Pro | **6.27** | **24.18** | **7.16** | **23.58** | 2.69 | 6.87 |
| Open | Mistral Large 2* | 3.58 | 6.40 | 4.48 | 10.45 | N/A | N/A |
|  | Command R+* | 4.18 | 8.96 | 4.78 | 7.46 | N/A | N/A |
|  | Llama 3.1 Instruct | 2.99 | 14.63 | 3.88 | 14.93 | 3.28 | 5.67 |

\* Due to budget constraints, we evaluate only Llama on RCA-agent among open-source models.

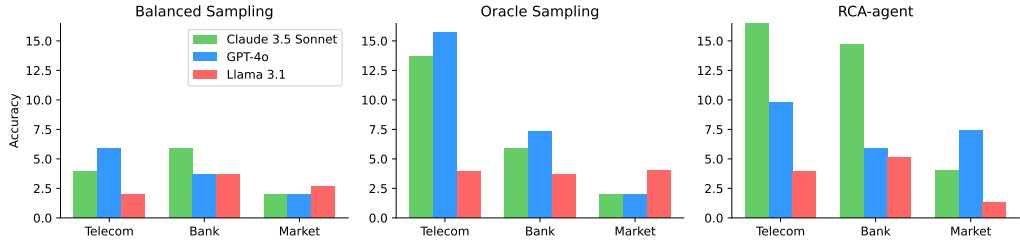

Figure 5: Accuracy distribution across three software systems (%).

only 11.34% accuracy even when using RCA-agent. A detailed discussion of these results is provided below. Due to space constraints, we focus our analysis on two proprietary models (Claude 3.5 and GPT-4o) and one open-source model (Llama 3.1) as representatives.

**Accuracy correlated with system complexity.** We observed that models consistently perform better on less complex systems. As shown in Figure 5, all models achieve higher accuracy on *Telecom* system compared to others. This is primarily due to the lower complexity of the *Telecom* data. As indicated in Table 1, the *Telecom* system has the smallest telemetry volume and significantly fewer unique root cause types, which simplifies the root cause analysis process. Moreover, it lacks the fault tolerance mechanisms used in *Bank* and *Market*, and has the fewest pods among systems. Thus, in Figure 5, Telecom's accuracy improved most significantly when using Oracle sampling and agent.

**Models struggle to identify multiple root cause elements.** We found that model performance declines significantly as the number of required root cause elements increases. As shown in Table 3, OpenRCA tasks were categorized into three levels: Easy (1 element), Mid (2 elements), and Hard (3 elements). Accuracy drops by at least half when the number of required elements increases from one to two. Furthermore, none of the current methods successfully resolve queries requiring all three

Table 3: Accuracy w.r.t task categorization, Easy (1 element), Mid (2 elements), Hard (3 elements)

| Model | Method | Category | | |
|---|---|---|---|---|
|  |  | Easy | Mid | Hard |
| Claude | Oracle | 8.72 | 3.50 | 0.00 |
|  | RCA-agent | 16.78 | 9.09 | 0.00 |
| GPT | Oracle | 9.40 | 4.90 | 0.00 |
|  | RCA-agent | 13.42 | 6.99 | 0.00 |
| Llama | Oracle | 7.38 | 1.40 | 0.00 |
|  | RCA-agent | 6.71 | 0.70 | 0.00 |

elements, where we provide a further analysis of these Hard queries in Appendix D. This finding suggests that most correctly answered queries by current models are relatively simple. Even with simpler queries, the models correctly solve only a small fraction of them. Additionally, it indicates that while OpenRCA is inherently challenging, it presents a clear difficulty gradient across tasks.

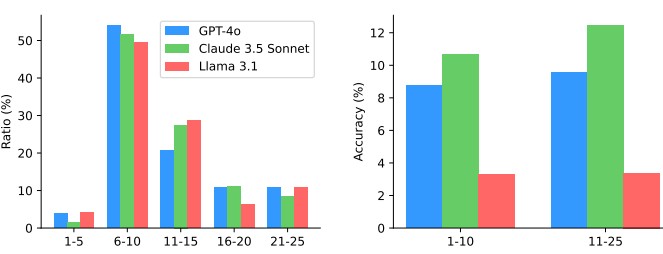

Table 4: Accuracy of answers involving failed execution. `Drop` indicates the percentage of accuracy reduction compared to Table 2

| Model | RCA-agent | |
| | Correct | Drop↓ |
| --- | --- | --- |
| Claude | **9.31** | 17.90 |
| GPT | 7.56 | 15.63 |
| Gemini | 0.85 | **68.40** |
| Llama | 3.18 | 18.04 |

Figure 6: **Left figure:** the reasoning length among all queries. **Right figure:** the accuracy of queries with different lengths.

**Agents prefer shorter reasoning but perform better with longer reasoning.** As shown in the left panel of Figure 6, about half of the responses fall within 10 steps, regardless of correctness, indicating that the agents tend to perform shorter reasoning. However, we also found that LLMs could perform better when they engage in longer reasoning. As shown in the right panel of Figure 6, models generally perform better when the reasoning length exceeds 10 steps. Combining both results, we hypothesize that current LLMs might be "lazy" in reasoning. Moreover, we observed an interesting finding with Claude. Although only a few of the model's responses are within 5 steps, it achieves a high accuracy of around 20%. To investigate the reason, we manually reviewed these cases, finding that they tend to be the simplest tasks without significant failure propagation and Claude properly resolved these cases with the short reasoning.

**Agent's performance is constrained by the model's error tolerance capability.** We found that a model's inability to effectively handle errors can significantly limit its performance as an RCA-agent. As shown in Table 2, while Gemini achieves the best performance among all sampling-based methods, its performance as an RCA-agent is the worst. Through a manual review of the agent's intermediate steps, we found that although all models may make mistakes when generating code, GPT and Claude can effectively utilize execution feedback, such as empty output or exception traceback, to revise their code or adjust their reasoning to bypass errors. In contrast, Gemini rarely does so. To quantify it, we further analyzed the accuracy of each model in cases involving failed executions. As shown in Table 4, under this setting, Gemini's accuracy sharply drops from its original accuracy of 2.69% to 0.85%, marking a significant reduction of 68.4%. In contrast, Claude and GPT exhibit only minimal declines of 15.6% and 17.9%, respectively. This suggests that the use of an RCA-agent adds an extra requirement to a model's capabilities, representing a trade-off: enhancing the scalability of language models for handling massive telemetry necessitates stronger error tolerance during reasoning.

## 6 CASE STUDY

We present a case generated by RCA-agent (Claude) to illustrate a potential direction to solve Open-RCA queries. As shown in Figure 7, this case is a query from the `bank` dataset, where the failure query requested the root cause component between 23:00 and 23:30 on March 6. To address this, the RCA-agent first analyzed metrics by extracting KPI sequences for the given time frame and calculating the P95 threshold to filter out anomalous data points. After identifying a large number of components with anomalies, the RCA-agent filtered out isolated noise points to reduce false positives. Components with persistent anomalies were considered genuinely affected by the failure. The RCA-agent then extracted traces involving the affected components, ranked the traces based on common call dependencies, and identified `Tomcat01` as the root cause due to its frequent self-calls in the downstream chain. Further log analysis pointed to network instability and memory leaks, leading the RCA-agent to conclude that packet loss was the failure cause. The agent correctly identified the root cause component required by the query, and the answer was deemed correct.

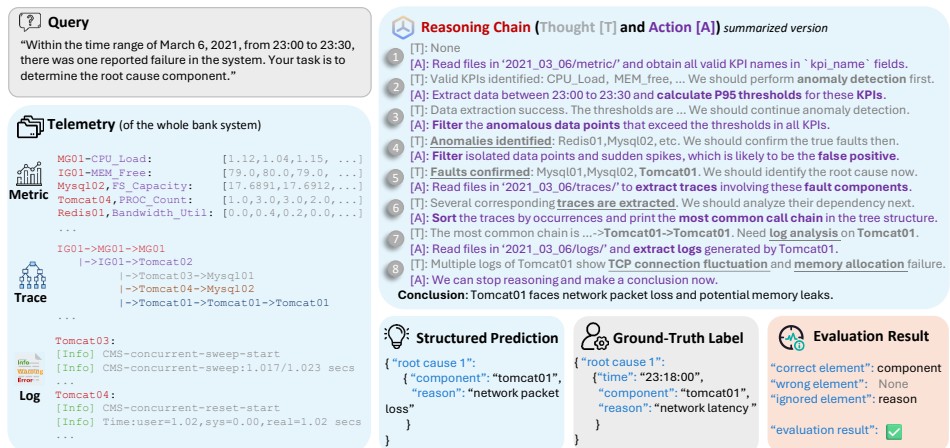

Figure 7: A case of RCA-agent (Claude) in `Bank` dataset. The reasoning chain is summarized.

# 7 RELATED WORK

**LLM for Real-world Software Engineering:** LLMs have demonstrated strength in various real-life application scenarios (Wang et al., 2025; 2024a), particularly within the SDLC: coding (Hong et al., 2024; Shi et al., 2024; Dinh et al., 2024; Li et al., 2025; Cao et al., 2025; Wan et al., 2024), testing (Chen et al., 2023; Pei et al., 2023; Wu et al., 2022; Xia et al., 2024b), logging (Xu et al., 2024a;c; Xie et al., 2024), and debugging (Chen et al., 2024a; Jimenez et al., 2024; Xia et al., 2024a; Xu et al., 2024b; Lee et al., 2024). However, the post-deployment stage has been largely overlooked. While the software researchers have recently acknowledged the potential of LLMs in this phase (Ahmed et al., 2023; Chen et al., 2024b), they mainly use LLMs as summarization tools for service failures, underutilizing their reasoning capabilities. Moreover, they rely on proprietary datasets and manual evaluation within corresponding organizations, limiting scalability. OpenRCA addresses this gap by introducing an open-access benchmark and evaluation framework for root cause analysis, leveraging LLMs' advanced reasoning abilities in post-deployment scenarios. This paves the way for further research and supports the automation of the entire SDLC.

**Root Cause Analysis (RCA):** The goal of the RCA task varies depending on the available telemetry and application scenario. If only metrics and service topology are available, the task is to locate the failure originating component (Yu et al., 2021) or its failure reason (Bi et al., 2024). If only logs are available, the goal is to identify the relevant logs generated when failure occurs (Amar & Rigby, 2019; Rosenberg & Moonen, 2020). If only traces are available, the goal is to identify the failure service operation (Yu et al., 2021). In scenarios where multiple telemetry types are available (Li et al., 2022b; Lee et al., 2023; Yu et al., 2023), the goal is various based on the needs (e.g., identifying the faulty component and the relevant KPIs). However, existing evaluation datasets (Li et al., 2022c; Lee et al., 2023; Yu et al., 2023) focus on single goals, providing labels for ad-hoc scenarios. OpenRCA addresses this by framing RCA as a goal-driven task, covering diverse scenarios of RCA.

# 8 DISCUSSION

**Limitation:** First, the failures collected by OpenRCA are all from distributed software systems. In the future, we aim to expand the dataset to include failures from systems with other architectures, such as monolithic systems, and use them to continuously update OpenRCA. Second, due to privacy and security concerns, OpenRCA is currently unable to gather first-hand failure reports from users or engineers. To approximate real-world scenarios, OpenRCA has synthesized queries based on common failure reports in a goal-driven manner. In the future, we hope to collaborate with companies to obtain real, anonymized failure reports for use as queries. Third, while our proposed RCA-agent makes models scalable for handling large volumes of telemetry without consuming extensive context, it increases the demands on the model's error tolerance capability. In the future, we aim to explore more robust methods to reduce these additional requirements for handling OpenRCA tasks.

**Conclusion**: We present OpenRCA, a benchmark and evaluation framework designed to assess language models' performance in real-world software engineering, which particularly fills the gap in post-deployment phase of software development lifecycle. By targeting the root cause analysis tasks, OpenRCA requires LLMs to perform advanced reasoning across diverse data types and structures. We also designed RCA-agent, an execution-based multi-agent system trying to solve the challenges in OpenRCA. The results expose current limitations in LLMs' ability to address these practical software tasks while opening new research opportunities in AI-driven software maintenance and reliability. As LLMs improve, success on OpenRCA could lead to significant advancements in automated troubleshooting and system reliability, potentially transforming the management of complex software systems.

## ETHICS STATEMENT

The OpenRCA is an open-sourced benchmark. Its telemetry data is sourced from the AIOps Challenge series AIOps (2018); Data under CC BY-NC 4.0, which permits the non-commercial use, redistribution, and adaptation of the material, provided appropriate credit is given and any derivatives are shared under identical license terms. The adaptations of OpenRCA to the original data are detailed in Sec. 2.4. No privacy or enterprise-sensitive information remains due to prior rigorous anonymization in the original challenges. The RCA-agent is strictly intended for academic benchmarking purposes. The RCA-agent, or any other LLM agent designed for solving OpenRCA tasks, should not be deployed in real-world production environments without a comprehensive safety assessment. Note that the authors hold sole legal responsibility for OpenRCA's data use, while users bear responsibility for any consequences of deploying RCA-agent in real-world scenarios.

## ACKNOWLEDGEMENTS

This paper was supported by the Guangdong Basic and Applied Basic Research Foundation (No. 2 024A1515010145) and the Shenzhen Science and Technology Program (No. ZDSYS20230626091 302006).

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

# A    BENCHMARK DETAILS

Table 5: Basic composition of datasets (with coarse-grained taxonomy)

| Dataset | Component level | Failure resource | Data type |
|---------|-----------------|------------------|-----------|
| Telecom | os,pod,db | cpu,net,db | trace,metric |
| Bank | pod | cpu,mem,io,net,jvm | log,trace,metric |
| Market | node,pod,service | cpu,mem,io,net,process | log,trace,metric |

## A.1    LICENSE

All telemetry data is originating from AIOps Challenge series AIOps (2018); Data, which is open-sourced and licensed under Creative Commons Attribution-Non Commercial 4.0 International: https://creativecommons.org/licenses/by-nc/4.0/.

## A.2    INTRODUCTION OF SERVICE SYSTEMS

**Telecom:** The telecom system consists of 22 virtual machine operating systems, 8 pods, 13 database services, 12 Redis middleware services, and multiple business services. The database services and Redis are directly deployed on the operating systems, while other business services run on pods hosted by the operating systems. The potential root cause components in this system include all operating systems, pods, and database services. There are five potential failure causes, categorized into CPU, network, and database failures. The telemetry data for the telecom system includes traces and metrics but excludes logs.

**Bank:** The bank system consists of 14 pods, 6 services, and multiple nodes. Five of these services are deployed across two pods each, while one service is deployed across four pods. This setup provides fault tolerance; when a pod hosting a service fails, another pod with the same service can handle the requests. The potential root cause components in this system include all 14 pods, with eight possible causes across five categories: CPU, memory, disk, network, and JVM failures. The telemetry data for the bank system includes metrics, traces, and logs.

**Market:** The market system comprises 6 server nodes, 40 pods, and 10 services. Each service is deployed across 4 different pods, similar to the bank system, providing fault tolerance. The potential root cause components in this system include all 6 nodes, 40 pods, and 10 services. Notably, if all pods hosting a particular service fail, this is considered a service-level failure rather than an individual pod failure. There are 15 possible causes of failure, spanning CPU, memory, disk, network, and process termination. The telemetry data for the market system includes metrics, traces, and logs.

Please note that all telemetry data is collected in the UTC+8 time zone. Therefore, when converting between timestamps and datetimes, ensure you specifically use the UTC+8 time zone.

## A.3    INTRODUCTION OF TELEMETRY

The telemetry data in OpenRCA encompasses metrics, logs, and traces, all structured in CSV files. Below is a brief introduction to each type of telemetry:

**Metrics:** Metric files contain time series data representing key performance indicators (KPIs) for different components. Each data point in a series is associated with a component (identified by `cmdb_id`) and a KPI type (`kpi_name`). By analyzing these metrics, one can detect anomalies in component performance. Below is an example of 10 rows from a metric file, which contains data points from different KPI:

```
timestamp,cmdb_id,kpi_name,value
1614787200,Tomcat04,OSLinux-CPU_CPU_CPUCpuUtil,26.2957
1614787200,Mysql02,Mysql-MySQL_3306_Innodb data pending writes,0.0
1614787200,Mysql02,Mysql-MySQL_3306_Innodb data pending reads,0.0
1614787200,MG01,OSLinux-CPU_CPU_CPUSysTime,0.3158
1614787200,MG01,OSLinux-CPU_CPU_CPUUserTime,25.5454
```

```
1614787200,Tomcat03,OSLinux-OSLinux_NETWORK_ens160_NETPacketsOut,115.0
1614787200,Tomcat03,OSLinux-OSLinux_NETWORK_ens160_NETPacketsIn,100.0
1614787200,Tomcat03,OSLinux-OSLinux_NETWORK_ens160_NETOutErr,0.0
1614787200,Redis02,redis-Redis_6379_Redis (latest_fork_usec),0.0
1614787200,Redis02,redis-Redis_6379_Redis (loading),0.0
```

All data points for a specific KPI of a component form a complete time series. For example, the `OSLinux-CPU_CPU_CPUCpuUtil` for `Tomcat03` can be represented as:

```
 timestamp value
1614787200 25.9228
1614787260 29.4098
1614787380 25.6756
1614787440 25.5445
1614787560 25.9666
1614787680 26.0624
1614787740 26.0624
1614787860 26.0859
1614787920 25.9044
1614788040 26.4985
...
```

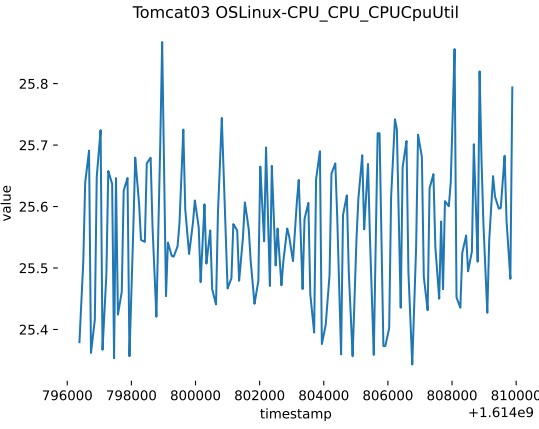

Figure 8: Time series of `OSLinux-CPU_CPU_CPUCpuUtil` for `Tomcat03`.

**Trace:** Trace files record the call chains between services, where each trace consists of multiple spans. A span represents a single communication event, capturing the interaction when one service makes a request to another. These spans are organized hierarchically, with each span having a parent span, which links it to the preceding span, forming a complete trace. By examining the relationships between spans within a trace, one can understand the dependencies between services and trace the path of potential failures across them. Below are 24 rows from a trace file, where each row represents a span, and together, they form a complete trace. (`trace_id`, `span_id`, `parent_id` are simplified to four digits)

```
timestamp,cmdb_id,parent_id,span_id,trace_id,duration
1614787515636,IG01,8603,8603,gw9703,80
1614787515636,IG01,8603,3432,gw9703,80
1614787517980,Tomcat02,3432,9209,gw9703,78
1614787517980,Tomcat02,9209,9210,gw9703,0
1614787517981,Tomcat02,9209,9211,gw9703,0
1614787517982,Tomcat02,9209,9212,gw9703,0
1614787517983,Tomcat02,9209,6004,gw9703,73
1614787518300,MG01,6004,6649,gw9703,71
1614787518300,MG01,6649,7505,gw9703,71
1614787200087,dockerB1,7505,6635,gw9703,66
1614787200089,dockerB1,6635,6636,gw9703,1
1614787200091,dockerB1,6635,6637,gw9703,1
1614787200094,dockerB1,6635,6638,gw9703,1
```

```
1614787200095,dockerB1,6635,6639,gw9703,0
1614787200112,dockerB1,6635,6640,gw9703,1
1614787200119,dockerB1,6635,6641,gw9703,1
1614787200120,dockerB1,6635,6921,gw9703,11
1614787518339,MG01,6921,6605,gw9703,8
1614787518339,MG01,6605,7506,gw9703,7
1614787200133,dockerB1,6635,6642,gw9703,1
1614787200136,dockerB1,6635,6643,gw9703,1
1614787200151,dockerB1,6635,6644,gw9703,1
1614787517983,Tomcat02,9209,9213,gw9703,0
1614787518058,Tomcat02,9209,9214,gw9703,0
```

Specifically, the call chain can be represented as a tree structure:

```
IG01, Timestamp:1614787515636
   IG01, Timestamp:1614787515636
      Tomcat02, Timestamp:1614787517980
         Tomcat02, Timestamp:1614787517980
         Tomcat02, Timestamp:1614787517981
         Tomcat02, Timestamp:1614787517982
         Tomcat02, Timestamp:1614787517983
            MG01, Timestamp:1614787518300
              MG01, Timestamp:1614787518300
                 dockerB1, Timestamp:1614787200087
                    dockerB1, Timestamp:1614787200089
                    dockerB1, Timestamp:1614787200091
                    dockerB1, Timestamp:1614787200094
                    dockerB1, Timestamp:1614787200095
                    dockerB1, Timestamp:1614787200112
                    dockerB1, Timestamp:1614787200119
                    dockerB1, Timestamp:1614787200120
                       MG01, Timestamp:1614787518339
                         MG01, Timestamp:1614787518339
                    dockerB1, Timestamp:1614787200133
                    dockerB1, Timestamp:1614787200136
                    dockerB1, Timestamp:1614787200151
         Tomcat02, Timestamp:1614787517983
         Tomcat02, Timestamp:1614787518058
```

It can also be represented as a dependency graph:

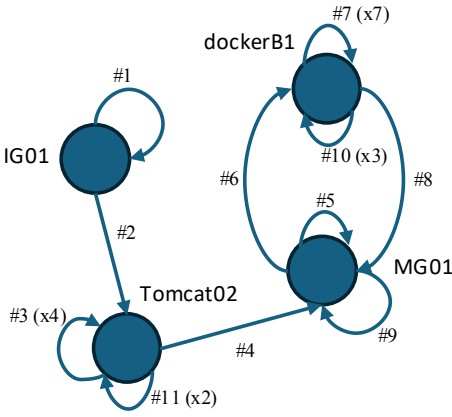

Figure 9: Dependency graph of the trace example, with `(xN)` indicating N self-requests.

**Log:** Log files capture runtime messages from components, often providing insight into the internal state or behavior of a system. Each log entry consists of a timestamp, a verbosity level, and a message. Logs are particularly useful for understanding the details of service operations and diagnosing issues. Below are 10 rows of log entries representing messages from different components: (`log_id` are simplified to four digits)

```
log_id,timestamp,cmdb_id,log_name,value
c763,1614787201,Tomcat01,gc,[CMS-concurrent-mark-start]
cfd3,1614787202,Tomcat01,gc,[CMS-concurrent-mark: 1.623/1.628 secs]
c87b,1614787202,Tomcat01,gc,[CMS-concurrent-preclean-start]
edbf,1614787202,Tomcat01,gc,[Times: user=0.02 sys=0.00, real=0.01 secs]
e319,1614787202,Tomcat01,gc,[CMS-concurrent-abortable-preclean-start]
6461,1614792889,Tomcat02,gc,[CMS-concurrent-preclean-start]
be53,1614792889,Tomcat02,gc,[CMS-concurrent-preclean: 0.015/0.015 secs]
ff38,1614792889,Tomcat02,gc,[CMS-concurrent-abortable-preclean-start]
9959,1614792894,Tomcat02,gc,[Times: user=1.21 sys=0.05, real=5.08 secs]
01a4,1614792894,Tomcat02,gc,[CMS-concurrent-sweep-start]
```

## A.4 Introduction of Root Cause Analysis

Failures in service systems are typically triggered by anomalies in specific components (e.g., nodes, containers) due to issues like disk saturation. These anomalies can propagate through service interactions—for instance, a full disk on a node may cause all services on containers deployed on that node to become unresponsive. This unresponsiveness then affects other services relying on them, leading to broader system failures at the application level. Timely and effective root cause localization is therefore a critical issue in software engineering to maintain software service system stability.

However, current research of RCA does not follow a unified task definition of the task output, typically due to the different requirements of specific scenarios. For example, if engineers only wonder which container is failure and want to redirect the traffic to other container, it does not need to know the in-depth reason of failure, i.e., only the failure components are needed. Thus, OpenRCA is considered to construct the practice of RCA into a goal-driven manner, where each goal refers to a combination of three key elements of the root cause, i.e., the failure originating component, occurrence time, and reason.

## A.5 Details of Data Calibration

We hired three SREs with over three years of RCA experience to calibrate the data. We provided the hired engineers with a standardized procedure to perform individual data calibration and verification. Specifically, they first extracted all telemetry data generated by the root cause component within 30 minutes before and after the labeled failure event. Next, they visualized KPIs for the relevant resource types based on the description of the root cause since we found such type-related KPIs (e.g., CPU, memory, network, disk I/O) are the most primary evidence for verification. In most cases, we can solely rely on type-related KPIs to verify the root cause time/component/reason. For example, if the failure was due to high CPU usage in a container, they examined all CPU-related KPIs within the time window. The engineers then identified anomalous data points in these KPIs, such as values exceeding the mean ± 3 standard deviations. However, in some instances where type-related KPIs do not show explicit anomalies, we use general business KPIs ("*golden signals*" like rr, sr, mrt) along with logs and traces to support the verifications. For example, in the Market dataset, we used business KPIs (mrt) along with trace latency and proxy logs to verify network packet corruption failures when packet transmit KPIs do not show explicit anomalies. In the Bank dataset, we combined business KPIs (mrt) along with service logs to validate high memory usage failures when memory usage KPIs do not show explicit anomalies. We do not verify telemetry for non-root cause components. Even though root cause components may not exhibit clear anomalies while their downstream components (affected by propagated faults) show significant anomalies, people still cannot precisely determine the exact root cause elements (e.g., the exact time of occurrence) and can only infer or guess a range of potential answers. Therefore, records without clear anomalies on root cause component were removed. Additionally, we checked for consistency between the labeled failure time and the actual onset of the root cause. As shown in Figure 10, Figure 11, and Figure 12, the failure record will be removed if the nearest data point of the telemetry around the failure occurrence time is not the first data point of an anomalous data duration. Finally, any failure records lacking telemetry data for the relevant time period or components were also filtered out. After completing individual calibration, the engineers cross-validated each other's results. Out of the original 412 cases, they disagreed on only 8 (less than 2%). Following thorough discussion, they reached a consensus on all cases.

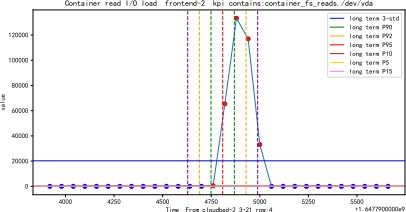

Figure 10: A case where the failure record matches the telemetry. Based on the record's guidance, engineers visualized the KPI `container_fs_reads` for the root cause component `frontend-2` within a 30-minute window, since the record suggests that the failure reason is `container read I/O load`. The red dashed line at the x-axis center indicates the failure occurrence time provided by the record, which is also the exact start time of the KPI spike. Given that the records align with direct evidence of the root cause, i.e., the corresponding KPI, we consider it possible to identify the root cause. Thus, this record is retained.

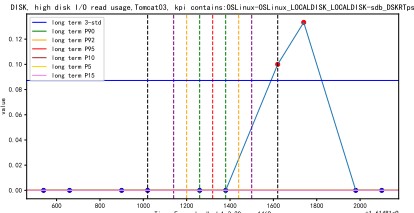 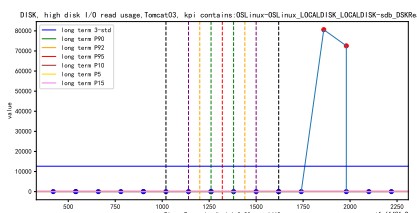

Figure 11: A case where the failure record does not match the telemetry. The engineers visualized all KPIs related to the disc since the record illustrates the failure reason is `high disc I/O read usage`. However, the exact failure occurrence time significantly deviates from the time provided in the failure records, as the nearest data point around the failure occurrence time is a normal point. Thus, this record is removed from our dataset.

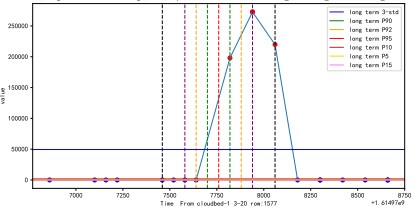 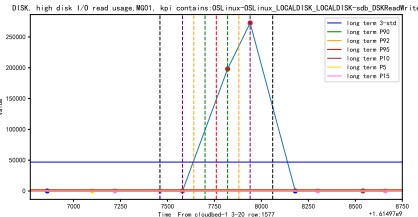

Figure 12: A case where the failure record matches the telemetry. The engineers visualized all KPIs related to the disc since the record illustrates the failure reason is `high disc I/O read usage`. The exact failure occurrence time is aligned with the label, as the nearest data point around the failure occurrence time is the first anomalous data point among an anomalous duration. Thus, the record is deemed accurate and is retained.

### A.6 DETAILS OF PROBLEM SYNTHESIS

To adapt to the different outputs of RCA, we use GPT-4o to synthesize the RCA problems for each failure record. We randomly select one task for each failure record, and then send the failure records with the corresponding task specifications to the model for query synthesis. A task specification is a JSON object that determines the input and output information of the query to be synthesized. The detailed system prompt and corresponding task specifications used for program synthesis are provided below.

**System Prompt**

```
Your task is to generate an issue related to DevOps failure diagnosis
    based on a given set of specifications. The goal is to make the issue
     realistic enough that an engineer could encounter at work.

The specifications provided to you include the following components:

'''known
(The known information explicitly provided in the issue.)
'''

'''query
(The target query that required the user to answer.)
'''

Your response should follow the JSON format below:

'''issue
(The generated issue based on the specifications.)
'''

For example:

{IN-CONTEXT EXAMPLES}

Some rules to follow:

1. Do not tell the user "how to solve the issue" (e.g., retrieve the
    telemetry data like metrics/logs/traces).
...

Now, let's get started!
```

**User Prompt:**

```
Please generate an issue related to DevOps failure diagnosis based on the
     following specifications:

'''known
{input_specification}
'''

'''query
{output_specification}
'''
```

**Task Specification:**

```
{
   "task_1": {
      "input": [
         "time range: {time_period}",
         "number of failures: {num}"
      ],
      "output": [
```

```
                "root cause occurrence time: {datetime}"
            ]
        },
        "task_2": {
            "input": [
                "time range: {time_period}",
                "number of failures: {num}"
            ],
            "output": [
                "root cause reason: {reason}"
            ]
        },
        "task_3": {
            "input": [
                "time range: {time_period}",
                "number of failures: {num}"
            ],
            "output": [
                "root cause component: {component}"
            ]
        },
        "task_4": {
            "input": [
                "time range: {time_period}",
                "number of failures: {num}"
            ],
            "output": [
                "root cause occurrence time: {datetime}",
                "root cause reason: {reason}"
            ]
        },
        "task_5": {
            "input": [
                "time range: {time_period}",
                "number of failures: {num}"
            ],
            "output": [
                "root cause occurrence time: {datetime}",
                "root cause component: {component}"
            ]
        },
        "task_6": {
            "input": [
                "time range: {time_period}",
                "number of failures: {num}"
            ],
            "output": [
                "root cause component: {component}",
                "root cause reason: {reason}"
            ]
        },
        "task_7": {
            "input": [
                "time range: {time_period}",
                "number of failures: {num}"
            ],
            "output": [
                "root cause component: {component}",
                "root cause occurrence time: {datetime}",
                "root cause reason: {reason}"
            ]
        }
    }
}
```

## A.7 DETAILS OF EVALUATION

OpenRCA requires all the methods to structure their final answer in the following JSON format:

```
{
    "1": {
        "root cause occurrence datetime": (A time in '%Y-%m-%d %H:%M:%S'
            format),
        "root cause component": (A component selected from the given '
            possible root cause component'),
        "root cause reason": (A reason selected from the given 'possible
            root cause reason'),
    },
    ...
}
```

During evaluation, the LLMs are tasked with correctly answering all the required elements of all the failures that occurred within the given time duration. If the LLM provides an answer to an element that was not required, that element will not affect the correctness of the answer. OpenRCA only focuses on whether the required elements are answered correctly and ignores the unnecessary fields generated by the model. All possible root cause components and reasons for each system is provided in each method's prompts. The prompt of these methods are generally discussed in Appendix B.2 and C.2.

## B  RCA-AGENT DETAILS

### B.1  FEATURES OF RCA-AGENT

**Scalability:** RCA-agent is not constrained by telemetry volume. By loading telemetry into memory via code execution rather than the LLM's context, it can process large datasets as long as memory allows. Expanding memory is far easier than increasing LLM context, ensuring scalability.

**Clarity:** RCA-agent avoids overwhelming the LLM with processing data analysis on large amounts of numbers, codes, and symbols in telemetry. By handling data analysis through code execution, the LLM can focus solely on the reasoning and decision-making process for the execution results.

**Efficiency:** Since telemetry is not directly fed into the LLM, it minimizes unnecessary token usage, keeping the context length concise and reducing overhead from irrelevant data.

**Generalizability:** RCA-agent does not require domain-specific knowledge beyond its telemetry schema. Instead, it only follows two general guidelines of root cause diagnosis. This allows RCA-agent to generalize effectively across various service systems.

### B.2  AGENT PROMPTS

We provide the system prompts of the Controller and Executor here for reference.

**Controller system prompt:**

```
You are the Administrator of a DevOps Assistant system for failure
    diagnosis. To solve each given issue, you should iteratively instruct
     an Executor to write and execute Python code for data analysis on
    telemetry files of target system. By analyzing the execution results,
     you should approximate the answer step-by-step.

There is some domain knowledge for you:

{BACKGROUND KNOWLEDGE OF SYSTEM}

## RULES OF FAILURE DIAGNOSIS:

What you SHOULD do:

1. **Follow the workflow of 'preprocess -> anomaly detection -> fault
    identification -> root cause localization' for failure diagnosis.**
...

What you SHOULD NOT do:

1. DO NOT include any programming language in your response.
...

The issue you are going to solve is:

{PROBLEM TO SOLVE}

Solve the issue step-by-step. In each step, your response should follow
    the JSON format below:

{
    "analysis": (Your analysis of the code execution result from Executor
        in the last step, with detailed reasoning of 'what have been done'
         and 'what can be derived'. Respond 'None' if it is the first step
        .),
    "completed": ("True" if you believe the issue is resolved, and an
        answer can be derived in the 'instruction' field. Otherwise "False
        "),
    "instruction": (Your instruction for the Executor to perform via code
        execution in the next step. Do not involve complex multi-step
```

```
        instruction. Keep your instruction atomic, with clear request of '
        what to do' and 'how to do'. Respond a summary by yourself if you
        believe the issue is resolved.)
}

Let's begin.
```

**Executor System Prompt**

```
You are a DevOps assistant for writing Python code to answer DevOps
    questions. For each question, you need to write Python code to solve
    it by retrieving and processing telemetry data of the target system.
    Your generated Python code will be automatically submitted to a
    IPython Kernel. The execution result output in IPython Kernel will be
     used as the answer to the question.

## RULES OF PYTHON CODE WRITING:

1. Reuse variables as much as possible for execution efficiency since the
    IPython Kernel is stateful, i.e., variables define in previous steps
    can be used in subsequent steps.
...

There is some domain knowledge for you:

{BACKGROUND KNOWLEDGE OF SYSTEM}

Your response should follow the Python block format below:

```python
(YOUR CODE HERE)
```
```

**Summary Prompt**

Note that once the Controller believe the task is completed, a summary prompts will be provided to controller for summarizing and structuring its answer to the JSON format required by OpenRCA:

```
Now, you have decided to finish your reasoning process. You should now
    provide the final answer to the issue. The candidates of possible
    root cause components and reasons are provided to you. The root cause
     components and reasons must be selected from the provided candidates
     .

{BACKGROUND KNOWLEDGE OF SYSTEM}

Recall the issue is: {PROBLEM TO SOLVE}

Please first review your previous reasoning process to infer an exact
    answer of the issue. Then, summarize your final answer of the root
    causes using the following JSON format at the end of your response:

{OPENRCA ANSWER FORMAT}
```

### B.3   BACKGROUND PROMPTS

We also designed three background prompts to introduce the schema of telemetry, i.e., Telecom, Bank, Market, and the possible failure components and reasons in the system. Note that these prompts are provided not only to the RCA-agent but also included in the prompts for sampling-based methods to provide basic system knowledge. All background prompts for all systems follow the format below:

```
## DATA SCHEMA
```

```
1. **Metric Files**:
{METRIC FILE SCHEMA}

2. **Trace Files**:
{TRACE FILE SCHEMA}

2. **Log Files**:
{LOG FILE SCHEMA}

## POSSIBLE ROOT CAUSE REASONS:
{FAILURE REASONS}

## POSSIBLE ROOT CAUSE COMPONENTS:
{FAILURE COMPONENTS}
```

# C EXPERIMENTAL SETUP

## C.1 CLARIFICATION

**Why not retrieval-based methods:** Unlike tasks such as code generation or summarization, which can leverage natural features (e.g., class names, function names, or keywords) for retrieval-augmented generation (RAG) Lewis et al. (2020), RCA faces challenges in identifying effective retrieval strategies due to the absence of such features in telemetry data. Actually, identifying faulty telemetry is a key challenge in RCA, as failures typically occur without clear indicators pointing to specific KPIs, failure logs, or anomalous call chains. To address this, we employed common sampling strategies from traditional RCA methods to construct our baseline.

**Why not chain-of-thought:** We did not explicitly instruct LLMs to perform chain-of-thought (CoT) reasoning Wei et al. (2022), as it generally resulted in poorer performance in our repetitive experiments. Table 6 compares the effect of explicitly requiring CoT versus not doing so. The results consistently show that CoT underperformed compared to prompts that did not require it. After manually reviewing both settings, we found that CoT often led models to focus on a few obvious anomalies listed in its thought, overlooking the given diagnostic guidance to explore the deeper failure propagation chain among these anomalies. We also report a brief case study in Appendix C.3.

Table 6: Repetitive comparison between our original prompt (Original) and CoT prompt (CoT) (%)

| GPT-4o | Balanced | | Oracle | |
|---|---|---|---|---|
| | Original | CoT | Original | CoT |
| Try-1 | 3.28 | 2.39 | 6.27 | 4.48 |
| Try-2 | 3.28 | **2.99** | **6.27** | **5.37** |
| Try-3 | **3.58** | 2.69 | 5.37 | 5.37 |
| Median | **3.58** | 2.99 | **6.27** | 5.37 |

**Selected Language Models:** The models in our experiments were accessed via APIs, with the open-source models using services from Mistral, Cohere, and Together.AI, as shown in Table 7. Due to differences in tokenizers, the open-source models often generated more tokens than GPT-4o, even with the same 128K token limit, causing some prompts to exceed the context window. To ensure an accurate evaluation, we reduced the number of KPIs in the prompt by eliminating interdependent KPIs in the oracle setting while keeping the total KPI count consistent between the oracle and sampling settings. Additionally, we selected the 70B version of Llama3.1 rather than 405B version since Together.AI does not support 128K context in 405B version.

Table 7: Checkpoint of each model

| Name | Checkpoint |
|---|---|
| Claude 3.5 | `claude-3-5-sonnet-20240620` |
| GPT-4o | `gpt-4o-20240513` |
| Gemini 1.5 Pro | `gemini-1.5-pro-exp-0801` |
| Mistral Large 2 | `mistral-large-instruct-2407` |
| Command R+ | `command-r-plus-08-2024` |
| Llama 3.1 Instruct | `meta-llama-70B-instruct` |

**Prompt of sampling-based methods:** Despite KPI sampling, original telemetry files contain redundant columns. We carefully filtered out irrelevant columns (e.g., component hash IDs) and compressed meaningful, long-encoded fields.

**Evaluation for sampling-based methods:** Since the sampling interval is limited to one minute, OpenRCA considers a failure time prediction correct if it falls within a one-minute window. We also manually verified that telemetry from oracle sampling still reveals the root cause components and failure reasons after sampling. Despite KPI sampling, original telemetry files contain redundant columns.

### C.2 PROMPTS OF SAMPLING-BASED METHODS

**Original Prompts**:

We first provide the prompt template used for both oracle sampling and balanced sampling in our experiment:

```
You will be provided with some telemetry data and an issue statement
    explaining a root cause analysis problem to resolve.

{BACKGROUND KNOWLEDGE OF THE SYSTEM}

{SAMPLED TELEMETRY DATA}

Now, I need you to provide an root cause analysis to the following
    question:

{PROBLEM TO SOLVE}

Note: A root cause is the fundamental factor that triggers a service
    system failure, causing other system components to exhibit various
    anomalous behaviors. It consists of three elements: the root cause
    component, the start time of the root cause occurrence, and the
    reason for its occurrence. The objective of root cause analysis may
    vary, aiming to identify one or more of these elements based on the
    issue. Each failure has only one root cause. However, sometimes a
    system's abnormal state may be due to multiple simultaneous failures,
     each with its own root cause. If you find that there is a call
    relationship between multiple components exhibiting abnormal behavior
    , these anomalies originate from the same failure, with the component
     at the downstream end of the call chain being the root cause
    component. The anomalies in the other components are caused by the
    failure. If there is no call relationship between the abnormal
    components, each component may be the root cause of a different
    failure. Typically, the number of failures occurring within half an
    hour does not exceed three.

Your response should be structured into a JSON format, itemising the
    relevant root cause information you find. You only need to provide
    the elements asked by the issue, and ommited the other fields in the
    JSON. The overall format is as follows:

{OPENRCA ANSWER FORMAT}

Please follow the format above to provide your response of current issue.

Response below:
```

In this prompt, we provide the background knowledge of each system same as what we did for RCA-agent. In addition, we also summarized the methodology to perform root cause analysis from the system prompt of RCA-agent (i.e., "Note that ...").

**CoT Prompts**:

We also provide the CoT prompts used in our repetitive experiment discussed in Appendix C.1.

```
You will be provided with some telemetry data and an issue statement
    explaining a root cause analysis problem to resolve.

{BACKGROUND KNOWLEDGE OF THE SYSTEM}

{SAMPLED TELEMETRY DATA}

Now, I need you to provide an root cause analysis to the following
    question:
```

```
{PROBLEM TO SOLVE}

Note: A root cause is the fundamental factor that triggers a service
    system failure, causing other system components to exhibit various
    anomalous behaviors. It consists of three elements: the root cause
    component, the start time of the root cause occurrence, and the
    reason for its occurrence. The objective of root cause analysis may
    vary, aiming to identify one or more of these elements based on the
    issue. Each failure has only one root cause. However, sometimes a
    system's abnormal state may be due to multiple simultaneous failures,
     each with its own root cause. If you find that there is a call
    relationship between multiple components exhibiting abnormal behavior
    , these anomalies originate from the same failure, with the component
     at the downstream end of the call chain being the root cause
    component. The anomalies in the other components are caused by the
    failure. If there is no call relationship between the abnormal
    components, each component may be the root cause of a different
    failure. Typically, the number of failures occurring within half an
    hour does not exceed three.

Please first conduct a comprehensive analysis on the given telemetry data
     step-by-step in your response. Then, summarize your findings using
    the following JSON format to provide a concise answer to the given
    issue at the end of your response. In the summarized answewr, you only
     need to provide the elements asked by the issue, and ommited the
    other fields in the JSON. The overall format is as follows:

{OPENRCA ANSWER FORMAT}

Please follow the format above to provide your response of current issue.

Response below:
```

The only difference between the original prompt and the CoT prompt is that the CoT prompt explicitly asked LLMs to perform a step-by-step analysis of the telemetry data.

## C.3 Case Study of CoT Prompting

We provide a case study to discuss the possible reason that the vanilla CoT does not work in RCA. In this case, we use Oracle sampling settings, and the model is GPT-4o. The RCA problem is:

> On March 21, 2022, between 12:00 and 12:30, the cloud service system, cloudbed-1, experienced a fault. The exact component responsible for this fault is currently unknown. Please identify the root cause component.

When using the original prompt, GPT-4o provides the correct answer, which is:

```
{
    "1": {
        "root cause component": "productcatalogservice",
        "root cause reason": "container read I/O load"
    }
}
```

However, when using CoT prompt, the full response is:

```
### Step-by-Step Analysis

#### 1. Identify the Time Window and Cluster
- The issue occurred on March 21, 2022, between 12:00 and 12:30.
- The affected cluster is 'cloudbed-1'.

#### 2. Analyze Service Metrics
- **Service Metrics**: Check for anomalies in 'metric_service.csv' for
    the specified time window.
```

```
- **Frontend Service**:
  - `mrt` (Mean Response Time) spikes significantly during the time window
    .
  - `mrt` values: 918.47, 13879.44, 13133.99, 10519.49, 13002.95, 12387.8,
    13131.03, 5234.86.
- **Checkout Service**:
  - `mrt` values: 398.92, 2375.0, 6340.0, 4821.43, 5500.0, 4950.0, 5075.0,
    3331.5.
- **Recommendation Service**:
  - `mrt` values: 64.81, 376.63, 416.07, 395.0, 395.0, 390.0, 395.0,
    161.54.
- **Product Catalog Service**:
  - `mrt` values: 360.68, 1851.71, 1886.56, 1824.71, 1837.86, 1837.15,
    1839.85, 573.48.

#### 3. Analyze Container Metrics
- **Frontend Containers**:
  - `frontend-0`, `frontend-1`, `frontend-2`, `frontend2-0` show high CPU
    usage and network receive metrics.
- **Checkout Containers**:
  - `checkoutservice-0`, `checkoutservice-1`, `checkoutservice-2`, `
    checkoutservice2-0` show high CPU usage and network receive metrics.
- **Recommendation Containers**:
  - `recommendationservice-0`, `recommendationservice-1`, `
    recommendationservice-2`, `recommendationservice2-0` show high CPU
    usage and network receive metrics.
- **Product Catalog Containers**:
  - `productcatalogservice-0`, `productcatalogservice-1`, `
    productcatalogservice-2`, `productcatalogservice2-0` show high CPU
    usage and network receive metrics.

#### 4. Analyze Node Metrics
- **Node Metrics**:
  - `node-1`, `node-2`, `node-3`, `node-4`, `node-5`, `node-6` show high
    CPU usage and disk usage metrics.

#### 5. Analyze Trace Data
- **Trace Data**:
  - Multiple traces show high latency in `frontend`, `checkoutservice`, `
    recommendationservice`, and `productcatalogservice`.

#### 6. Analyze Log Data
- **Log Data**:
  - Logs from `frontend`, `checkoutservice`, `recommendationservice`, and
    `productcatalogservice` show errors and high latency.

### Summary of Findings
- The root cause component is likely to be `frontend` service due to its
    significant impact on `mrt` and high CPU and network usage.
- The anomalies in `checkoutservice`, `recommendationservice`, and `
    productcatalogservice` are likely propagated from the `frontend`
    service.

### JSON Output
```json
{
    "1": {
        "root cause component": "frontend"
    }
}
```
```

While LLMs with CoT prompting also identify multiple anomalous data points across the frontend, checkoutservice, recommendationservice, productcatalogservice in step 5, it directly determines the

service with the most significant latency, frontend, as the root cause component, instead of following the guidance to analyze traces and identify the failure's originating component. We consider this may be because the model became lost in the massive context and complex reasoning process among heterogeneous telemetry data, leading to focusing only on superficial findings from its earlier analysis in stage 2. Therefore, to better reflect the model's upper bound in solving OpenRCA, we did not use CoT prompting as a baseline.

## D SUPPLEMENTARY ANALYSIS

As shown in Table 3, all large language models (LLMs), even in the agent setting, achieve 0% accuracy on Hard queries. To further investigate which element contributes to the difficulty of solving Hard queries, we conducted an in-depth analysis of the relative difficulty among different elements.

Our approach is to evaluate the *partially correct accuracy* (i.e., `Partial` in Table 2) for each element across all failure cases to infer their relative difficulty. Note that directly using the *absolute value* of each element's accuracy is insufficient, as LLMs may occasionally guess an element correctly by chance (this is also the reason we place less emphasis on partially correct results when calculating accuracy in Sec. 5). To better reflect difficulty, we should first exclude the impact of random guessing by calculating the improvement over random accuracy. Specifically, we compute the difference (delta) between each method's absolute accuracy and the random guessing accuracy. The element with the smallest delta is the hardest to predict, as the method cannot improve upon its random guessing accuracy. Below, we compute the random guessing accuracy for each element:

- **Reason**: As shown in Table 1, the number of reasons varies across systems. The random guessing accuracy is calculated as $(51/5 + 136/8 + 148/15)/335 = 11.06\%$.
- **Component**: Similarly, the number of components differs by system. The random guessing accuracy is $(51/15 + 136/14 + 148/44)/335 = 4.92\%$.
- **Time**: As detailed in Appendix A.5 and Sec. 4.1, each query involves identifying the root cause within a 30-minute window, with predictions considered correct if they fall within one minute of the actual failure time. The random guessing accuracy is $1/30 = 3.33\%$.

The absolute accuracy and delta values of each element's accuracy are provided below.

Table 8: Comparison of partially correct accuracy across elements. R, C, and T refer to Reason, Component, and Time, respectively.

| Type | Category | Absolute | | | Delta | | |
|---|---|---|---|---|---|---|---|
| | | **R.** | **C.** | **T.** | **R.** | **C.** | **T.** |
| **Oracle** | Claude | 12.56 | 13.54 | 11.30 | +1.50 | **+8.62** | +7.97 |
| | GPT-4o | 13.39 | 12.56 | 12.14 | +2.33 | +7.64 | **+8.81** |
| | Llama | 11.30 | 11.72 | 6.70 | +0.24 | **+6.80** | +3.37 |
| **Agent** | Claude | 19.67 | 18.00 | 14.23 | +8.61 | **+13.08** | +10.90 |
| | GPT-4o | 18.00 | 14.65 | 13.39 | +6.94 | +9.73 | **+10.06** |
| | Llama | 7.53 | 3.35 | 2.51 | -3.53 | -1.57 | **-0.82** |
| **Random** | - | 11.06 | 4.92 | 3.33 | N/A | N/A | N/A |

The table shows that regardless of the method used, identifying the **reason** is consistently the most challenging task, while **component** and **time** are comparatively easier to determine. Specifically, Claude performs better at identifying components than time, whereas GPT exhibits the opposite trend. Llama, limited by its weaker coding and mathematical capabilities, frequently encounters execution failures during reasoning, often preventing it from completing the task successfully. This observation aligns with the findings in Table 3.

