# OpenReview forum: "OpenRCA: Can Large Language Models Locate the Root Cause of Software Failures?"
_ICLR.cc/2025/Conference — ICLR 2025 Poster_

### Official Review · Reviewer_RqHx · 2024-11-01

**Soundness:** 3
**Presentation:** 3
**Contribution:** 2
**Rating:** 5
**Confidence:** 3

**Summary:**

The paper preposes OpenRCA, a benchmark for root cause analysis that requires models to extract potential root cause from large quantities of telemetry data, logs and traces. The paper also proposes RCAagent as a potential method to solve this root cause analysis problem since vanilla models struggle. The paper uses three SOTA closed-source models and three open-source models for the evaluation on OpenRCA.

**Strengths:**

- RCA remains an important problem in realistic scenarios.
- Good paper structure, explains the key problem, terminologies, and ideas very clearly.
- Clean and good dataset with clear structures, etc.
- A realistic RCA benchmark, with huge effort made towards ensuring fairness and denoising.

**Weaknesses:**

### Major
- The “golden set” identified in the oracle sampling might be not realistic enough. In real-life tasks, it is also one of the key challenge for the model to understand and select the correct KPI in order to diagnose the problem.
- Using agent has been a widely adopted method in other research fields. This paper fails to provide any insight into the nature of the problem and merely uses a general planner-executor agent framework almost without applying any domain knowledge. The only domain knowledge is a 42-line long rule given to the planner, which basically outlines all the things that the LLM should do. As a result the agent part seems really redundant and unnecessary. The same argument applies to the case study.
- There is no point trying vanilla LLM on this task. The author acknowledges in the paper that it is virtually impossible to give the telemetry data to the LLM, which is why they adopt the sampling method. There are two flaws in this: 1. LLM is not designed to process huge amount of data, so an agent evaluation/previous baseline evaluation should be more proper, and 2. There is no tell that if the performance of the LLM is bounded by the sampling.

### Misc
- Some apparent spelling mistakes, like “Partitial” in table 2, or “Accuray” in table 3
- The instruction given to the baseline in the artifact is far less comprehensive than the one given to the agent. Maybe fairer to give more domain knowledge to the baseline models?
- The CSV file for the record in the artifact is confusing and consists lots of excess newlines?
- Rather than reducing the size of the benchmark, why don’t do a success rate calculation on each system first and then compute the final score on the three success rates?

**Questions:**

- Does oracle sample generalize to general RCA tasks? Is it a realistic baseline?
- Does the 42-line rule in the system prompt given to the planner LLM generalize to other cases as well? What is the insight in designing an agent this way specifically for this task?
- Is the benchmark intended for LLM or agents? Why wasn't other more specialized tools/agents from previous work tested and reported on the benchmark?
It would also be great for the authors to address the misc weaknesses.

---

> ### Author Response · Authors · 2024-11-21
>
> Thank you for your time and effort towards this review. Before addressing each question individually, we would like to first provide the following general clarifications:
>
> **General Clarifications:**
>
> 1. **OpenRCA benchmark designed to evaluate whether LLMs can assist in solving goal-driven RCA tasks, no matter what specific LLM-based methods are used** (see **Q10**). We hope OpenRCA will inspire impactful tools for post-development phases, much like Cursor and Copilot support development tasks (Sec. 1).
> 2. **Sampling strategies are widely used and straightforward for RCA to reduce the redundancy of large volume of telemetry data** [1,2,3]. Before proposing more complex solutions, it is crucial to analyze the performance **upper bound** of this baseline to highlight the challenges of OpenRCA tasks. Thus, we adopts the **oracle setting**, a common strategy to test the best performance of LLMs on software tasks that require information-lossy filtering (e.g.,sampling or retrieval) to manage large volumes of data [4]. The suboptimal performance of oracle sampling baseline underscores the task’s complexity and the need for more advanced solutions like agents. (See **Q1** and **Q3**)
> 3. **RCA-Agent serves as another baseline to explore potential directions for enhancing LLM performance in RCA tasks**. Its design is driven by the insight that RCA tasks require integrating general RCA workflow knowledge to process extensive non-natural language tokens containing heterogeneous information from various data sources, such as logs, traces, and metrics (Sec. 3). Moreover, directly inputting data into the LLM is not scalable, as the telemetry data typically varies significantly across different systems. **Therefore, we select execution-based agents as baseline rather than other general-structured baselines like RAG-based agent, since non-natural language data can be processed more effectively and scalably by the corresponding data analytics library in the execution kernel than by the LLM itself**. (See **Q2**)
>
> ---
>
> The detailed responses for each question are provided below:
>
> **Q1**: *The “golden set” identified in the oracle sampling might be not realistic enough. In real-life tasks, it is also one of the key challenge for the model to understand and select the correct KPI in order to diagnose the problem.*
>
> **A1**:  **The oracle sampling setting is *intentionally designed* as an idealized and unrealistic baseline to investigate the *upper bound* of sampling-based methods**. Sampling is a common strategy for handling the large volume of telemetry data in traditional RCA methods [1,2,3], as telemetry often contains significant redundancy. Thus, we first explore whether LLMs can directly solve OpenRCA tasks using sampling. However, as discussed in Sec. 5 (lines 310–313), even with ground-truth information, the oracle sampling setting fails to perform well on OpenRCA tasks. This highlights the need and motivation for developing RCA-Agent as a baseline for OpenRCA.
>
> To avoid potential misunderstandings, we have clarified the rationale behind selecting sampling-based methods and using the oracle setting in detail in Sec. 4.1 of the paper.

---

> ### Author Response · Authors · 2024-11-21
>
> **Q2**: *Using agent has been a widely adopted method in other research fields. This paper fails to provide any insight into the nature of the problem and merely uses a general planner-executor agent framework almost without applying any domain knowledge. The only domain knowledge is a 42-line long rule given to the planner, which basically outlines all the things that the LLM should do. As a result the agent part seems really redundant and unnecessary. The same argument applies to the case study.*
>
> **A2**: We would like to answer this question by discussing (1) our insight, (2) the domain knowledge in the agent, and (3) the necessity of the agent.
>
> 1. **About the insight**:
>
>     As discussed in Sec. 3, the core challenge for LLMs lies in the large volume of heterogeneous telemetry (log, trace, metrics) associated with each query, which predominantly consists of non-natural language data, such as numbers, GUIDs, and error codes. While LLMs struggle to process such data, agents can handle it effectively with the support of an executable Python environment. Specifically, an agent can use execution environment to retrieve, process, and summarize the data in a unified manner. For example, it can process telemetry to extract and abstract key features (e.g., mean and standard deviation of time-series data, structured trace graphs, log abstractions with error/fatal verbosity) and then feeds these summaries into the LLM, reducing the need for the models to handle extensive context for multiple complex tasks. This approach significantly reduces token consumption while maintaining a global view of the data.
>
>     Notably, other reviewers (e.g., 41p3) acknowledged RCA-Agent’s potential to inspire future research. As a baseline, RCA-Agent offers a strategy to improve accuracy in OpenRCA tasks and serves as a foundation for further exploration.
>
> 2. **About the domain knowledge in the agent**:
>
>     To enable RCA-Agent to read and analyze telemetry data while reasoning about RCA problems with generalizability, the only domain knowledge provided is the *basic workflow* followed by human IT operators (Sec. 3): *anomaly detection → fault identification → root cause localization*, along with the *general order of analysis*: *metrics → traces → logs*. These high-level guidelines offer broad suggestions and essential context (e.g., what may or may not be a root cause) without dictating specific implementation details. That said, the LLM independently determines how to perform detailed operations, such as selecting parameters for time-series metric analysis, strategies for identifying root causes among multiple anomalous components using traces, methods for isolating relevant logs, or algorithms for integrating data to summarize the final root cause.  The LLM can also determine whether additional analyses are needed or if prior steps should be revisited to refine conclusions. This design balances providing sufficient domain knowledge to guide the Agent while allowing the LLM flexibility to adapt its approach to different systems and datasets by its own decision.
>
> 3. **About the necessity of the agent**:
>
>     We consider the RCA-Agent is needed due to two reasons:
>
>     1. LLMs struggle to process extensive non-natural language tokens containing heterogeneous information from various data sources. While straightforward sampling strategies can alleviate some issues, they are not scalable due to significant variations in telemetry data across systems. An execution-based agent effectively addresses these challenges by leveraging Python libraries to handle different types of heterogeneous data and utilizing memory for storing extensive context. Compared to scaling up the LLM context window, scaling up the computer memory is far more cost-efficient and practical, making it well-suited for OpenRCA tasks.
>     2. The 42-line general guideline is insufficient for a vanilla LLM to complete the RCA process, as it lacks the detailed implementation capabilities required. For example, metric analysis requires statistical operations on each KPI time-series data, which LLMs struggle to individually perform but can easily accomplish through data analytics libraries. Similarly, trace analysis involves retrieving and aggregating spans into a tree or graph structure to trace root causes across the whole call chain, which LLMs struggle to individually perform but can easily accomplish through database libraries. These gaps highlight the need for an execution-based agent as a baseline for OpenRCA.

---

> ### Author Response · Authors · 2024-11-21
>
> **Q3**: *There is no point trying vanilla LLM on this task. The author acknowledges in the paper that it is virtually impossible to give the  telemetry data to the LLM, which is why they adopt the sampling method.  There are two flaws in this: 1. LLM is not designed to process huge  amount of data, so an agent evaluation/previous baseline evaluation should be more proper, and 2. There is no tell that if the performance  of the LLM is bounded by the sampling.*
>
> **A3**: We would like to answer this question by discussing (1) the motivation for vanilla LLM baselines, (2) the agents and previous baselines, and (3) whether LLM is bounded by the sampling.
>
> 1. **About vanilla LLM with sampling**:
>
>     As discussed in our answer to Q1, sampling is a common strategy in traditional RCA to reduce data volume. If a straightforward and general sampling strategy could solve OpenRCA problems with LLMs, then the problem is likely not complex enough to necessitate an agent. Our goal is to provide a comprehensive evaluation of potential LLM-based methods for OpenRCA tasks, so it is essential to include sampling-based LLM prompting baselines.
>
> 2. **About Agent/previous baselines**:
>
>     It is challenging to determine whether the sampling-based method or the agent performs better without experimental comparison. If OpenRCA's data and tasks are simple, sampling-based methods might outperform because errors among multiple agents can propagate and cascade [8]. To provide rigorous guidance for future research and a comprehensive understanding, we consider both straightforward methods—sampling-based and agent—as baselines.
>
>     Additionally, OpenRCA is a goal-driven task designed to assess whether LLMs can solve RCA problems. While traditional baselines could theoretically be adapted to OpenRCA through data and task reformulation, their system-specific nature necessitates additional training sets for training from scratch. Since OpenRCA does not provide these additional training sets, adapting previous baselines is challenging. Consequently, these baselines fall outside the scope of our approach.
>
> 3. **About whether LLM is bounded by the sampling**:
>
>     For RCA tasks, both sampling-based prompting and agent approaches involve data filtering to ensure scalability with large data volumes, which can theoretically lead to some information loss (i.e., both methods might be “bounded”). To address this, we included the *oracle* sampling method as a baseline, which incorporates ground truth during the filtering process to measure the potential upper limit of sampling methods. In software engineering, where data volumes are large, such information loss is common. Therefore, oracle settings are often used to assess the upper bound performance of a vanilla LLM baseline with information-lossy filtering methods (e.g., *oracle retrieval* for retrieval-based baselines [4]).
>
> **Q4**: *Some apparent spelling mistakes, like “Partitial” in table 2, or “Accuray” in table 3*
>
> **A4**: Thanks for suggestion. We have revised our paper accordingly.
>
> **Q5**: *The instruction given to the baseline in the artifact is far less comprehensive than the one given to the agent. Maybe fairer to give more domain knowledge to the baseline models?*
>
> **A5**: The domain knowledge in all three baselines (two sampling methods and one agent) is similar, focusing on the basic RCA workflow. However, the agent requires more extensive specific instructions for code execution and reasoning loops. For instance, the agent must understand how to detect anomalies with Python, when to stop reasoning, and the restriction against writing or caching local files to prevent altering the file system and disrupting the process. In contrast, the sampling-based methods do not involve such complex procedures, so they do not require these additional prompts.
>
> **Q6**: *The CSV file for the record in the artifact is confusing and consists lots of excess newlines?*
>
> **A6**: We reviewed all artifacts and found that each query ends with a newline, which results in seemingly extra newlines in all `query.csv` files. However, these newlines are normal and do not affect the reading of the CSV files or the execution of experiments. Specifically, CSV files enclose text with newlines in quotation marks to indicate that the multi-line text in the text editor (e.g., vscode) is actually a single row of data.
>
> **Q7**: *Rather than reducing the size of the benchmark, why don’t do a success rate calculation on each system first and then compute the final score  on the three success rates?*
>
> **A7**: We believe that with balanced data, both macro accuracy (which is suggested in the question) and micro accuracy (which is adopted in OpenRCA) can be effectively utilized if needed. Additionally, controlling the amount of evaluation data aids in detailed label verification and labeling with less human effort, while also reducing LLM time/money costs when evaluating new methods on OpenRCA.

---

> ### Author Response · Authors · 2024-11-21
>
> **Q8**: *Does oracle sample generalize to general RCA tasks? Is it a realistic baseline?*
>
> **A8**: The sampling strategy is common in RCA methods. The oracle sampling method is unrealistic as discussed in Sec.4.1 line 282. It is intentionally designed to be idealistic to evaluate the performance upper bound of sampling-based methods.
>
> More details can be found in our answer to Q1.
>
> **Q9**: *Does the 42-line rule in the system prompt given to the planner LLM generalize to other cases as well? What is the insight in designing an agent this way specifically for this task?*
>
> **A9**: Yes. As illustrated in the result of Sec.5, RCA-Agent performance is generalizable across three systems. The insight is to use execution environment to process large volume of non-natural language context with general RCA domain knowledge guidance for the basic reasoning workflow, rather than feed these context to the models directly.
>
> More details can be found in our answer to Q2.
>
> **Q10**: *Is the benchmark intended for LLM or agents? Why wasn't other more  specialized tools/agents from previous work tested and reported on the  benchmark?  It would also be great for the authors to address the misc weaknesses.*
>
> **A10**: OpenRCA focuses on whether LLMs can handle RCA tasks, regardless of the specific method used (prompting, agents, etc.). In software engineering scenarios, agents are not necessarily superior to well-constructed prompting methods. For example, on the program repair tasks [4,5], carefully designed prompting methods [6] might outperform complex agent methods [7]. Therefore, OpenRCA is not a benchmark designed for a specific method, but rather for evaluating the ability of any LLM-based method to address the real-world challenge of RCA.
>
> Since there are no other valid agents for RCA or other post-development tasks (existing agents focus on development tasks like coding) when basic sampling-based prompting methods are verified to perform poorly via our experiments, we have to to propose RCA-Agent as another baseline to provide a potential direction for solving OpenRCA queries. Furthermore, we do not consider traditional RCA tools or baselines because (1) they involve different and ad-hoc task formulations (Sec.2.3), while OpenRCA adopts a more general goal-driven task formulation in a broader scope, and (2) OpenRCA focuses on the evaluation, so it does not contain system-specific training data required by the traditional tools.
>
> Additionally, we address misc issues in sections A4-A7.
>
> More details can be found in our answer to Q3.
>
> **Reference**
>
> [1] Huang H, Zhang X, Chen P, et al. Trastrainer: Adaptive sampling for distributed traces with system runtime state[J]. Proceedings of the ACM on Software Engineering, 2024, 1(FSE): 473-493.
>
> [2] Chen Z, Jiang Z, Su Y, et al. TraceMesh: Scalable and Streaming Sampling for Distributed Traces[J]. arXiv preprint arXiv:2406.06975, 2024.
>
> [3] He S, Feng B, Li L, et al. STEAM: observability-preserving trace sampling[C]//Proceedings of the 31st ACM Joint European Software Engineering Conference and Symposium on the Foundations of Software Engineering. 2023: 1750-1761.
>
> [4] Jimenez, Carlos E., et al. "SWE-bench: Can Language Models Resolve Real-world Github Issues?." *The Twelfth International Conference on Learning Representations* (ICLR) 2024.
>
> [5] https://www.swebench.com/
>
> [6] Xia, Chunqiu Steven, et al. "Agentless: Demystifying llm-based software engineering agents." *arXiv preprint arXiv:2407.01489* (2024).
>
> [7] Yang, John, et al. "Swe-agent: Agent-computer interfaces enable automated software engineering." *arXiv preprint arXiv:2405.15793* (2024).
>
> [8] Guo, Taicheng, et al. "Large language model based multi-agents: A survey of progress and challenges." *arXiv preprint arXiv:2402.01680* (2024).

---

> ### Author Response · Authors · 2024-11-24
> **Additional Experiment Results for Q5**
>
> **Additional Experiment Results for Q5:**
>
> As noted in our initial response to Q5, the prompts for both sampling methods and the agent convey similar core information (the basic RCA workflow). To address concerns about the potential impact of prompt detail on experimental fairness, we replaced the sampling methods’ original prompt with the agent’s prompt and conducted oracle sampling experiments using GPT-4o and Claude 3.5. The table below compares the accuracy of the original oracle methods with that of the alternative prompt:
>
> | **Model** | **Original Accuracy** | **Alternative Accuracy** |
> | --- | --- | --- |
> | Claude 3.5 | 5.37 | 3.88 |
> | GPT-4o | 6.27 | 4.48 |
>
> The results show that replacing the prompt actually reduced performance, likely because the agent's longer, more detailed instructions are difficult for single-prompt models to process effectively. Moreover, many high-level guidelines in the agent's prompt are specifically designed to guide code execution and the agent's main loop, which are unnecessary for sampling-based methods. Actually, the original prompt was already optimized for sampling-based methods through extensive testing. Thus, we consider our original comparison is fair.
>
> We have also updated the artifact to include the alternative prompt script, available at `baseline/DirectLM_detailed.py`.

---

> ### Author Response · Authors · 2024-11-24
>
> As we near the conclusion of the discussion period, we would like to ensure that our responses have addressed the questions raised in your initial reviews. We are confident that we have addressed the original concerns (major and misc) in our response and are eager to continue the discussion if there are any additional questions. If you find our responses satisfactory, we kindly request the reviewer to consider raising the score.
>
> Thank you.

---

> ### Author Response · Authors · 2024-11-29
> **Kind Reminder**
>
> Dear Reviewer RqHx,
>
> As the extended discussion period approaches its conclusion, we hope our responses have adequately addressed the questions and concerns raised in your initial review. We believe that we have addressed both **major** and **misc** concerns in our response and would be happy to continue the discussion if needed. If you find our responses satisfactory, we kindly request the reviewer to consider adjusting the rating.
>
> Sincerely,
>
> Authors

---

### Official Review · Reviewer_41p3 · 2024-11-03

**Soundness:** 3
**Presentation:** 4
**Contribution:** 3
**Rating:** 8
**Confidence:** 4

**Summary:**

This paper provides OpenRCA, a benchmark set for root cause analysis for software systems. The benchmark set is composed of 335 failures and the task is to identify the timing of the failure along with the component that failed and its reason. The paper also proposed RCA-agent, a multi-agent approach that utilizes a coe execution environment to effectively utilize telemetry data. While the best performing LLM was able to solve 6% of failure cases, RCA-agent increased it 11%.

**Strengths:**

The paper introduces a novel benchmark dataset for root cause analysis. The benchmark set is carefully designed with emphasis on consistency and quality. This in my opinion is a significant contribution as it can likely help advance the field, motivating further contributions utilizing this dataset. The second contribution on RCA-agent is limited in its originality as multi-agent systems utilizing code execution are widely available. Nevertheless, showcasing the increase in accuracy through an agentic approach motivates further research on this. The paper is very well written and organized.

**Weaknesses:**

The paper puts emphasis on creating a high quality dataset and one of the processing steps was the removal of failure records where the root cause cannot be identified using the given telemetry. Envisioning a system which acts based on the analysis from automated root cause analysis, it is vital to be able to flag if the root cause cannot be identified from given data. Therefore, the benchmark set should reflect that in my opinion.

Due to the highly stochastic nature of agentic systems as well as LLMs, I think evaluations would be stronger if authors can provide error bars.

As the paper focuses on root cause analysis for distributed systems, one insight that would make the paper stronger is how LLMs and the proposed RCA-agent perform as the scale of the system increases. That would help the readers understand limitations and applicability of such approaches.

**Questions:**

- The comparative analysis of LLMs are interesting but given the rather low performance of individual LLM’s, I’m curious to see if one would get any additional boost in performance when they are used in an ensemble fashion. This also helps to understand if the LLMs considered identify similar failures or not.
- Table 3. For the hard category, all LLM’s even in the agent setting gets 0. I’m guessing maybe identifying the reason is harder than identifying the timing of the issue. Following this intuition, I’m wondering if there is a task that is difficult across the board?
- Figure 5. The accuracy distribution for RCA-agent is completely different than the sampling based ones. For instance, using GPT-4o, we see that the accuracy is reduced for Telecom and bank datasets but increased for Market dataset. Similarly, using Llama3.1, we see that the accuracy is similar for Telecom and Bank but reduced for the Market dataset. I’d love to hear more insights on these behaviors.
- Could you clarify which model size you used for Llama3.1-instruct?
- Nit: Table 4. Explain what drop ratio is in the title.

---

> ### Author Response · Authors · 2024-11-21
>
> Thank you so much for your encouraging comments and insightful questions and suggestions. Please refer to our clarifications below:
>
> **Q1**: *The paper puts emphasis on creating a high-quality dataset and one of the processing steps was the removal of failure records where the root cause cannot be identified using the given telemetry. Envisioning a system which acts based on the analysis from automated root cause analysis, it is vital to be able to flag if the root cause cannot be identified from given data. Therefore, the benchmark set should reflect that in my opinion.*
>
> **A1**: Thank you for your thoughtful question. We initially considered adding some special queries to test whether LLMs could correctly determine when no failure exists and respond with “no root cause can be identified.” However, after careful consideration, we chose to focus the RCA task on its primary objective: identifying the root cause when evidence of failure is present. This aligns with standard workflows, where anomaly detection is handled as an upstream task to confirm the existence of a failure. Consequently, all our failure cases assume the presence of an identifiable root cause. We agree, however, that exploring end-to-end RCA tasks, including anomaly detection, could be valuable and may consider adding such systems in future updates.
>
> **Q2**: *Due to the highly stochastic nature of agentic systems as well as LLMs, I think evaluations would be stronger if authors can provide error bars.*
>
> **A2**: Considering the runtime and token costs, we run experiments for each settings three times and then report the median results Sec.5. Due to the limited number of repetitions, we present the raw results of the oracle and agent below instead of using error bars.
>
> |  | **test-1** | **test-2** | **test-3** | **median** |
> | --- | --- | --- | --- | --- |
> | Claude (Oracle) | 5.37 | 4.78 | 5.67 | 5.37 |
> | GPT-4o (Oracle) | 5.67 | 6.27 | 6.57 | 6.27 |
> | Llama (Oracle) | 3.58 | 3.88 | 3.88 | 3.88 |
> | Claude (Agent) | 11.64 | 10.45 | 11.34 | 11.34 |
> | GPT-4o (Agent) | 8.36 | 9.25 | 8.96 | 8.96 |
> | Llama (Agent) | 3.28 | 3.58 | 3.28 | 3.28 |
>
> **Q3**: *As the paper focuses on root cause analysis for distributed systems, one insight that would make the paper stronger is how LLMs and the proposed RCA-agent perform as the scale of the system increases. That would help the readers understand limitations and applicability of such approaches.*
>
> **A3**: We appreciate the reviewers' insightful suggestions. Establishing systems of varying scales indeed helps in assessing the scalability of different methods, which not only changes data volume but also complicates failure propagation patterns across components, making the problem more challenging for all RCA methods. However, deploying additional components and capturing more failures on current system datasets is challenging, requiring months for deployment. In future updates, we will consider designing sub-datasets with different scales within the same system to evaluate method scalability if time and resources permit.
>
> **Q4**: *The comparative analysis of LLMs are interesting but given the rather low performance of individual LLM’s, I’m curious to see if one would get any additional boost in performance when they are used in an ensemble fashion. This also helps to understand if the LLMs considered identify similar failures or not.*
>
> **A4**: We employed a majority voting strategy to ensemble the predictions of the three models (Claude, GPT, and Llama). The results are as follows:
>
> |  | **Oracle** | **Agent** |
> | --- | --- | --- |
> | Majority voting | 4.78 | 3.88 |
>
> It can be observed that both the oracle and the agent exhibit lower accuracy compared to solely using the best-performing LLM. This suggests that different models typically do not identify the same failure root causes. Moreover, we observed that the agent's accuracy during the ensemble process is lower than that of the oracle. We believe this discrepancy partly stems from the inherent randomness of the agent, whereas oracle sampling does not involve such stochasticity. As a result, different agents may identify different correct root causes, which could lead to this variation.

---

> ### Author Response · Authors · 2024-11-21
>
> **Q5**: *Table 3. For the hard category, all LLM’s even in the agent setting gets 0. I’m guessing maybe identifying the reason is harder than identifying the timing of the issue. Following this intuition, I’m wondering if there is a task that is difficult across the board?*
>
> **A5**: Analyzing the relative difficulty of predicting different elements in RCA tasks is indeed an interesting question. One approach is to evaluate the *partially correct accuracy* for each element across all failure cases to infer their relative difficulty. However, directly using the *absolute value* of each element's accuracy is insufficient, as LLMs may occasionally guess an element correctly by chance (this is also the reason we place less emphasis on partially correct results when calculating accuracy in Sec.5). To better reflect difficulty, we should first exclude the impact of random guessing by calculating the improvement over random accuracy. Specifically, we compute the difference (delta) between each method's absolute accuracy and the random guessing accuracy. The element with the smallest delta is the hardest to predict, as the method cannot improve upon its random guessing accuracy. Below, we compute the random guessing accuracy for each element:
>
> 1. **Reason**: As shown in Table 1, the number of reasons varies across systems. The random guessing accuracy is calculated as (51/5 + 136/8 + 148/15)/335 = 11.06%.
> 2. **Component**: Similarly, the number of components differs by system. The random guessing accuracy is (51/15 + 136/14 + 148/44)/335 = 4.92%.
> 3. **Time**: As detailed in Appendix A.5 and Sec. 4.1, each query involves identifying the root cause within a 30-minute window, with predictions considered correct if they fall within one minute of the actual failure time. The random guessing accuracy is 1/30 = 3.33%.
>
> The absolute (abs) and delta value of each element's accuracy are provided below.
>
> |  | **reason (abs)** | **component (abs)** | **time (abs)** | **reason (delta)** | **component (delta)** | **time (delta)** |
> | --- | --- | --- | --- | --- | --- | --- |
> | Claude (Oracle) | 12.56 | 13.54 | 11.30 | +1.50 | **+8.62** | +7.97 |
> | GPT-4o (Oracle) | 13.39 | 12.56 | 12.14 | +2.33 | +7.64 | **+8.81** |
> | Llama (Oracle) | 11.30 | 11.72 | 6.70 | +0.24 | **+6.80** | +3.37 |
> | Claude (Agent) | 19.67 | 18.00 | 14.23 | +8.61 | **+13.08** | +10.90 |
> | GPT-4o (Agent) | 18.00 | 14.65 | 13.39 | +6.94 | +9.73 | **+10.06** |
> | Llama (Agent) | 7.53 | 3.35 | 2.51 | -3.53 | -1.57 | **-0.82** |
> | **random** | **11.06** | **4.92** | **3.33** |  |  |  |
>
> The table shows that regardless of the method used, identifying the **reason** is consistently the most challenging task, while **component** and **time** are comparatively easier to determine. Specifically, Claude performs better at identifying components than time, whereas GPT exhibits the opposite trend. Llama, limited by its weaker coding and mathematical capabilities, frequently encounters execution failures during reasoning, often preventing it from completing the task successfully. This observation aligns with the findings in Table 4.
>
> Overall, identifying the root cause **reason** remains the most difficult challenge.

---

> ### Author Response · Authors · 2024-11-21
>
> **Q6**: *Figure 5. The accuracy distribution for RCA-agent is completely different than the sampling based ones. For instance, using GPT-4o, we see that the accuracy is reduced for Telecom and bank datasets but increased for Market dataset. Similarly, using Llama3.1, we see that the accuracy is similar for Telecom and Bank but reduced for the Market dataset. I’d love to hear more insights on these behaviors.*
>
> **A6**: We appreciate the reviewer’s insightful question. We believe this issue relates to differences in the models' capabilities to process and understand various data types. As shown in Table 1 and Figure 2, the telemetry compositions of three systems differ significantly—Bank and Telecom have substantially more *traces* but far fewer *metrics* compared to Market.  To further analyze the reason, we manually analyzed 15 execution records each from GPT-4o and Llama3.1 on the three systems. Below, we will discuss how the models' different capabilities in *metric  analysis* and *trace analysis* affect the agent's performance.
>
> 1. We observed that GPT-4o rarely encountered execution failures during metric analysis (3 out of 15), whereas Llama3.1 frequently did (13 out of 15). Since LLMs typically identify anomalous KPIs by analyzing statistical features (std, mean, etc.) in time-series data, we attribute Llama3.1's lower performance on Market to poor-quality data analytics code and limited numerical reasoning capabilities. This observation aligns with the conclusion from the arena leaderboard, which indicates that Llama 3.1 performs much worse than GPT-4o on coding and math [1].
> 2. We also observed that during trace analysis, GPT-4o often exhibited hallucinations, misattributing the root cause to the most upstream component (usually the frontend) in the anomalous call chain, even when earlier metric analysis indicated this component was functioning normally (5 out of 15). Llama3.1 also showed a similar issue (6 out of 15). However, due to Llama3.1’s limited general capabilities on long-context processing [1], its oracle setting performed poorly on Bank and Telecom. Thus, despite occasional hallucinations, Llama3.1’s Agent still slightly outperformed its oracle baseline on Bank and Telecom.
>
> **Q7**: *Could you clarify which model size you used for Llama3.1-instruct?*
>
> **A7**: 70B. All model checkpoints are discussed in Appendix C.1 Table 6.
>
> **Q8**: *Nit: Table 4. Explain what drop ratio is in the title.*
>
> **A8**: We have revised our manuscript accordingly. Specifically, “Drop” indicates the percentage of accuracy reduction compared to Table 2.
>
> **Reference**
>
> [1] https://lmarena.ai/

---

> > ### Comment · Reviewer_41p3 · 2024-11-29
> >
> > Thank you authors for your response! I maintain my score of 8. I'd also recommend adding the task difficulty related discussion to the appendix as I found it very informative.

---

> > > ### Author Response · Authors · 2024-11-29
> > > **Thank you!**
> > >
> > > Thank you for your suggestions and supports! We will include the discussion of each element's difficulty into the appendix of our revised manuscript.

---

### Official Review · Reviewer_RSze · 2024-11-04

**Soundness:** 3
**Presentation:** 3
**Contribution:** 3
**Rating:** 6
**Confidence:** 3

**Summary:**

The paper introduces OpenRCA, an innovative benchmark designed to evaluate large language models in identifying root causes of software failures using diverse, long-context telemetry data. The authors evaluate state-of-the-art LLMs on this benchmark, offering a high-level analysis of their performance. The authors also introduce RCA-Agent as a tailored solution for OpenRCA task. The findings suggest significant room for enhancing LLM capabilities in solving real-world service reliable problems.

**Strengths:**

- OpenRCA pioneers the application of LLMs in the post-development phase of the software life-cycle, highlighting a new direction to use LLMs for identifying and analyzing real-world software issues.

- The authors’ rigorous data-cleaning process in building OpenRCA is valuable, involving system selection, data balancing, data calibration and query synthesis.

- The goal-driven task design covers various aspects of RCA tasks, framing them in a generalized, natural-language query format.

- The authors conduct a comprehensive experiments of popular closed and open LLMs with different strategies (Balanced Sampling, Oracle Sampling and RCA-Agent). And the authors also provide high-level comparative analysis for each setting based on experiment results.

**Weaknesses:**

- OpenRCA includes only three distributed software systems, which is small and may limit the benchmark’s generalizability. As noted by the authors, accuracy appears to correlate with system complexity, and the telemetry distribution varies across systems. Although OpenRCA provides 335 failure cases, the limited number of software systems could introduce significant variance.

- The benchmark does not fully disentangle the capabilities of long-input retrieval, non-natural language understanding, and software problem identifying. The long, complex, non-natural language input introduces additional difficulty for LLMs, potentially limiting performance beyond the task of software failure identifying. Adding an ablation study could help isolate the challenges introduced by long, non-natural language input.

- The data collection approach seems to highly reliant on human effort, which may cause difficulty in updating and scaling. Collecting software telemetry and manually annotating Oracle sampling seems to be labor-intensive. Actually, frequent updates are crucial for keeping LLM benchmarks quality, particularly in consideration of potential data leakage.

- As the authors note, the current benchmark includes only distributed systems. Expanding the benchmark to incorporate systems from other domains would provide a broader evaluation of LLM capabilities.

- Typo: In line 035, "post-development phrases" --> "post-development phases."

**Questions:**

- The authors categorize tasks into three levels: Easy, Mid, and Hard, based on the number of elements. Given that system complexity varies among repositories (for instance, Telecom has the smallest telemetry volume), could the authors further explain on why element count was chosen as the primary criterion for task difficulty? How did the authors account for the impact of system complexity on task difficulty?

- The authors state in Line 363 that "models generally follow the same trend: accuracy increases with more reasoning steps." However, Figure 6 does not appear to strongly support this, as accuracy varies significantly across different models and lengths. Could the authors provide further clarification or evidence on how this conclusion was reached?

---

> ### Author Response · Authors · 2024-11-21
>
> Thanks for your thorough review of the paper. Please refer to our clarifications below:
>
> **Q1**: *OpenRCA includes only three distributed software systems, which is small and may limit the benchmark’s generalizability. As noted by the authors, accuracy appears to correlate with system complexity, and the telemetry distribution varies across systems. Although OpenRCA provides 335 failure cases, the limited number of software systems could introduce significant variance.*
>
> **A1**:  The proprietary nature of telemetry and system failure data makes real-world datasets rare. Companies are often unwilling to share private data or invest in anonymization without clear incentives. Additionally, collecting data in the post-development phase for RCA  tasks is more challenging than in the development phase because it involves deploying software on devices, requiring numerous devices and considerable time. Therefore, unlike development tasks that typically cover around ten projects [1], OpenRCA, which collects data from three distinct systems, is comprehensive and diverse for post-development tasks, standing out among other RCA datasets. Specifically, Telecom represents a distributed cloud database architecture, while Bank and Market are microservice architectures. These systems embody diverse application scenarios and architectural diversity for evaluation. Thus, we believe these three systems have generalizability. We will also keep updating OpenRCA to include more systems as discussed in Sec. 8.
>
> Furthermore, it is commonly observed in other software task benchmarks [1,2,3] that the performance usually varies due to the different complexity of software projects (e.g., systems, repositories, etc.). These project differences help evaluate how methods perform across projects and highlight their characteristics and comparative advantages. Thus, the variance in model performance across systems does not necessarily imply a lack of data diversity or generalizability.
>
> Moreover, in OpenRCA, data diversity is determined not only by the number of systems but also by the unique features of each system. As shown in Tables 1 and 5, each system includes dozens of distinct components and root cause reasons, and different root cause reasons affecting different components typically exhibit entirely different patterns. For example, failures caused by upstream versus downstream services, or by CPU versus network issues, propagate differently and affect components in unique ways. Furthermore, as discussed in Appendix A, each system has unique component levels. Root causes at different levels, such as nodes versus pods, also exhibit distinct patterns. Thus, we consider OpenRCA cases to be still diverse.
>
> **Q2**: *The benchmark does not fully disentangle the capabilities of long-input retrieval, non-natural language understanding, and software problem identifying. The long, complex, non-natural language input introduces additional difficulty for LLMs, potentially limiting performance beyond the task of software failure identifying. Adding an ablation study could help isolate the challenges introduced by long, non-natural language input.*
>
> **A2**: OpenRCA is designed to evaluate multiple interrelated LLM capabilities in challenging real-world post-development software tasks. Similar to benchmarks for other real-world (development) software tasks [1], it is the complex and interdependent nature of these tasks makes them particularly challenging for LLMs. For instance, retrieving information from extensive telemetry requires LLMs to first have a basic understanding of telemetry (composed of non-natural language with rich software domain knowledge), and to know what information to retrieve, making it difficult to separate these capabilities.
>
> Moreover, there are numerous benchmarks for evaluating individual capabilities such as long-input retrieval, non-natural language understanding, and domain-specific knowledge. Therefore, OpenRCA aims to assess models' comprehensive abilities by treating RCA as a complex, integrated task, rather than evaluating each LLM capability independently.
>
> Additionally, OpenRCA task data is inherently heterogeneous and complex, involving non-natural language tokens, long contexts, and requiring specialized domain knowledge of software. Thus, isolating these features and separating the data for ablation studies are very challenging. However, if the reviewer has suggestions for specific ablation study settings, we would be happy to consider incorporating additional experiments.

---

> ### Author Response · Authors · 2024-11-21
>
> **Q3**: *The data collection approach seems to highly reliant on human effort, which may cause difficulty in updating and scaling. Collecting software telemetry and manually annotating Oracle sampling seems to be labor-intensive. Actually, frequent updates are crucial for keeping LLM  benchmarks quality, particularly in consideration of potential data leakage.*
>
> **A3**: The significant effort we have invested so far is due to the poor maintenance of existing offline data and the lack of standards for collecting online data during runtime. As a result, we have had to spend considerable time on cleaning and reconstruction to eliminate inconsistencies in schema and structure.  To this end, we present a well-defined task formulation and data organization that can be used to guide the observable tools and pipelines (e.g., Prometheus, SkyWalking, ELK Stack) to automatically collect historical failure data and labels from the online service systems in compliance with our standards (rather than manually reconstruct the offline data with inconsistent schema). Additionally, we provide a query synthesis framework for automatically generating query data when the telemetry data adheres to our schema. The only human involvement required is verifying label correctness, which is far less labor-intensive than creating labels from scratch and necessary for ensuring data quality. As a result, future data collection, calibration, and construction will require significantly less effort.
>
> Furthermore, the oracle sampling baseline is generated automatically using existing ground-truth labels if the failure and telemetry data/labels align with our standard.
>
> **Q4**: *As the authors note, the current benchmark includes only distributed systems. Expanding the benchmark to incorporate systems from other domains would provide a broader evaluation of LLM capabilities.*
>
> **A4**: Modern service systems predominantly use distributed architectures, particularly microservices, making it challenging and time-consuming to find partners still using traditional monolithic systems. To ensure our evaluation reflects mainstream architectures, the three systems we currently provide are all distributed: Bank and Market are microservice architecture, while Telecom represents a general distributed database. These systems also exhibit diversity within distributed systems, given the significant differences in their designs.
>
> We believe this selection sufficiently ensures diversity and scope. However, we acknowledge the importance of including monolithic systems and are actively seeking enterprise partners using such architectures to further expand our benchmark, as discussed in Sec. 8.
>
> **Q5**: *The authors categorize tasks into three levels: Easy, Mid, and Hard, based on the number of elements. Given that system complexity varies among repositories (for instance, Telecom has the smallest telemetry volume), could the authors further explain on why element count was chosen as the primary criterion for task difficulty? How did the authors account for the impact of system complexity on task difficulty?*
>
> **A5**: Element count (1, 2, 3) is the most *quantifiable* factor for task complexity. This is because identifying more elements requires more fine-grained analysis. For example, determining both the root cause component and reason is clearly more challenging than identifying the failure component alone. Furthermore, the element count (1, 2, 3) provides a straightforward and natural metric for assessing difficulty levels. Thus, we propose using element count as the primary criterion for task difficulty.
>
> In contrast, while tasks from different systems vary in difficulty, quantifying this precisely using system features, such as the number of components, telemetry volume, or reason types, is challenging. For instance, with GPT-4o, RCA-Agent and Oracle methods both perform best on Telecom, which has fewer components and less telemetry than the other systems. However, RCA-Agent outperforms Oracle on Market, while Oracle performs better on Bank. This suggests that system difficulty cannot be determined by a few features easily. For example, as shown in Table 1, Market has more unique root cause components and reasons, whereas Bank has higher telemetry volume. Given the complexity of system features and telemetry, it is difficult to quantify query difficulty solely by system attributes.

---

> ### Author Response · Authors · 2024-11-21
>
> **Q6**: *The authors state in Line 363 that "models generally follow the same trend: accuracy increases with more reasoning steps." However, Figure 6  does not appear to strongly support this, as accuracy varies significantly across different models and lengths. Could the authors  provide further clarification or evidence on how this conclusion was reached?*
>
> **A6**: We sincerely appreciate the reviewer for pointing this out, which might be also confusing to other readers. We would like to clarify that when analyzing the execution trajectories of the agent, we identified a *10-step* reasoning length as a key threshold: over half of the samples with reasoning lengths within 10 steps, but cases exceeding 10 steps generally show a slightly higher accuracy. Specifically, in some instances requiring longer reasoning steps, LLMs often use additional reasoning to reflect on intermediate conclusions produced in earlier steps. They then gather all available evidence to either support or refute their updated conclusions about root causes. For example, if the agent detects a conflict between intermediate results—like finding a potentially anomalous component through metrics while observing no abnormalities in traces or logs—it may re-execute the earlier metric analysis to avoid false positives. This process may lead to better overall performance.
>
> In the earlier version, we aimed to provide readers with a more fine-grained view by dividing reasoning lengths into intervals of 5 steps for the visualization. However, thanks to the reviewer’s feedback, we realized that the right sub-figure of Figure 6 exhibited less consistent growth across consecutive intervals due to typical fluctuations in reasoning length. For instance, when RCA-Agent fails to complete a step, it is allowed to re-execute that step up to three additional times, introducing possible fluctuations of 3–6 steps in reasoning length. To address this and avoid potential misinterpretation, we revised the right sub-figure of Figure 6 by grouping cases based on the 10-step threshold. The revised figure in our revised manuscript now better highlights the trends associated with this threshold. In addition to the figure in the revised manuscript, we have also provided the data from the figure in tabular form below for reference:
>
> |  | <10 | >10 |
> | --- | --- | --- |
> | GPT | 8.76 | 9.57 |
> | Claude | 10.67 | 12.46 |
> | Llama | 3.27 | 3.38 |
>
> We also agree that simply stating *"accuracy increases with more reasoning steps"* could be confusing. Therefore, we have revised the claim in our manuscript to indicate that "*samples with reasoning lengths exceeding 10 steps generally achieve slightly higher accuracy*."
>
> **Reference**
>
> [1] Jimenez, Carlos E., et al. "SWE-bench: Can Language Models Resolve Real-world Github Issues?." *The Twelfth International Conference on Learning Representations*.
>
> [2] Just, René, Darioush Jalali, and Michael D. Ernst. "Defects4J: A  database of existing faults to enable controlled testing studies for  Java programs." *Proceedings of the 2014 international symposium on software testing and analysis*.
>
> [3] Jiang, Zhihan, et al. "A large-scale evaluation for log parsing techniques: how far are we?." *Proceedings of the 33rd ACM SIGSOFT International Symposium on Software Testing and Analysis*. 2024.

---

> > ### Comment · Reviewer_RSze · 2024-11-25
> >
> > Thanks for the authors' explanation. It clarified many of my initial concerns. I appreciate the authors' efforts in data collection and reconstruction. However, I still have reservations regarding the small repository size and the structured, lengthy nature of the input telemetry data. The SWE-bench, which you mentioned, includes a larger number of repositories to mitigate variance and uses a simpler input format, code and NL instructions, that aligns more closely with LLMs' capabilities.

---

> > > ### Author Response · Authors · 2024-12-02
> > > **Kind Reminder**
> > >
> > > Dear Reviewer RSze,
> > >
> > > As the extended discussion period approaches its conclusion, we hope our further clarifications have adequately addressed the concerns raised in your latest response. We have also included an extra table to highlight the **size** and **diversity** of OpenRCA compared to current datasets. If you find our responses satisfactory, we kindly request the reviewer to consider adjusting the rating.
> > >
> > > Sincerely,
> > >
> > > Authors

---

> > > > ### Comment · Reviewer_RSze · 2024-12-02
> > > >
> > > > Thank you for providing additional explanations and supplementary statistics. I improved my score.

---

> > > > > ### Author Response · Authors · 2024-12-02
> > > > > **Thank you!**
> > > > >
> > > > > We are glad that our clarifications addressed your concerns. Thank you again for your reviewing effort!

---

> ### Author Response · Authors · 2024-11-24
>
> As we near the conclusion of the discussion period, we would like to ensure that our responses have addressed the questions raised in your initial reviews. We would be happy to provide more clarifications if needed, or more experiments if there are suggestions for any specific settings. If you find our responses satisfactory, we kindly request the reviewer to consider raising the score.
>
> Thank you.

---

> ### Author Response · Authors · 2024-11-25
> **Further clarifications to project size and the nature of input**
>
> Thanks for your reply! We are happy to provide further clarifications regarding the project size & variance and the nature of input data in OpenRCA compared to SWE-bench:
>
> ## Summarization
>
> 1. The diversity and size is similar (**3** scenarios vs **4** scenarios, both include **300** test cases).
> 2. The lengthy and complex input is the **common nature** of software tasks (both involve over **100,000** tokens even using **oracle retrieval** in SWE-bench or **oracle sampling** in OpenRCA.
>
> Here are detailed clarifications:
>
> ### **Project Size and Variance**:
>
> 1. **Project-level Diversity**: The Python repositories of SWE-bench comes from 4 different scenarios: **Data Analytics** (scikit-learn, sympy, astropy, xarray), **Software** **Development** (django, pytest, sphinx, pylint, flask), **Visualization** (seaborn, matplotlib), **Socket** (requests). The accuracy within each scenario is similar: Data Analytics: 2.5%-5%, Software Development: 2.5%-5% (except for flask), Visualization: <2.5%, Socket: 15%. In contrast, all three systems from OpenRCA come from different scenarios: **Telecom**, **Bank**, and **Market**. Notably, as shown in our [benchmark data](https://drive.google.com/drive/u/2/folders/1wGiEnu4OkWrjPxfx5ZTROnU37-5UDoPM), the Market system is deployed across two different physical device groups, contributing 148 failure cases (70 and 78 cases, respectively). This demonstrates comparable diversity in application domains.
> 2. **Source Limitation**: SWE-bench’s repositories were curated from millions of open-source GitHub projects, whereas OpenRCA’s systems were selected from a more limited pool of fewer than ten available systems. In the absence of a suitable benchmark for post-development RCA evaluation, **OpenRCA represents the best achievable solution at present**. As discussed in **Q3**, we are actively exploring collaborations with more enterprises to expand the dataset following our established pipeline and standards.
> 3. **Case-level Diversity**: As discussed in **Q1**, data diversity is influenced not just by the number of systems but also by the variety of features within them. OpenRCA includes **73 unique root cause components** and **28 unique root cause reasons**, with propagation patterns varying widely across these elements. Meanwhile, SWE-bench’s most popular version, SWE-bench-lite, includes 300 issues—fewer than OpenRCA’s 335 failure cases. Thus, at the case level, OpenRCA demonstrates notable diversity.
>
> ### **Nature of Input**:
>
> 1. **Length**: SWE-bench baseline’s inputs are inherently lengthy (**>100,000 tokens** even with **oracle** **retrieval**, which is similar to our **oracle sampling** in OpenRCA), as they encompass issues related to an entire codebase. To reduce the input length, SWE-bench's baselines use retrieval-based strategy to filter out unrelated artifacts. Similarly, OpenRCA baselines leverage sampling-based methods to adapt telemetry inputs to LLM context limits, mirroring SWE-bench’s retrieval-based approach for handling large-scale artifacts. Lengthiness is a common characteristic of software data (e.g., code, documents, telemetry) across software scenarios. Hence, the input nature between SWE-bench and OpenRCA is not significantly different.
> 2. **Format and its Complexity**: OpenRCA baseline’s input is simple, consists of three parts: (1) **query** (NL), (2) **telemetry** (logs, traces, metrics), and (3) **structured output** (JSON). Similarly, SWE-bench uses three parts: (1) **issue** (NL), (2) **artifact** (documents, code files), and (3) **structured output** (Unix diff patch). Moreover, the structured nature is common in software artifacts. For example, the Unix diff patch used in SWE-bench is also semi-structured, like the logs used in OpenRCA. This shared characteristic also reflects common practices in software artifacts.
>
> We sincerely hope these clarifications can address the concerns in your reservations.

---

> ### Author Response · Authors · 2024-11-28
> **Further clarifications to the repository size and diversity**
>
> To address concerns about repository size and diversity, we would like to clarify that OpenRCA is the largest and most diverse RCA dataset that can be collected so far. Below, we compare OpenRCA with existing open-source evaluation datasets proposed in other RCA approach papers:
>
> | **name** | **publication** | **data source** | **system count** | **system type** | **data type** | **data size** | **case number** | **unique component** | **unique reason** |
> | --- | --- | --- | --- | --- | --- | --- | --- | --- | --- |
> | CIRCA [1] | KDD’22 | synthetic | 1 | DAG | metric | N/A | N/A | N/A | N/A |
> | PyRCA [2] | arXiv | synthetic | 1 | DAG | metric | N/A | N/A | N/A | N/A |
> | RCD [3] | NeurIPS’22 | collected | 1 | Microservice | metric | 16.40 MB | 50 | 13 | 2 |
> | MicroRCA [4] | NOMS’20 | collected | 1 | Microservice | metric | 23.70 KB | 95 | 13 | 3 |
> | MicroRank [5] | WWW’21 | collected | 1 | Microservice | trace | 16.24 MB | 150 | 10 | 4 |
> | PDiagnose [6] | ISPA’21 | collected | 2 | Microservice, Database | log, trace, metric | 1.35 GB | 99 | 29 | 7 |
> | Eadro [7] | ICSE’23 | collected | 2 | Microservice | log, trace, metric | 1.22 GB | 234 | 38 | 3 |
> | Nezha [8] | ESEC/FSE’23 | collected | 2 | Microservice | log, trace, metric | 2.97 GB | 101 | 35 | 5 |
> | **OpenRCA** | N/A | **collected** | **3** | **Microservice,  Database** | **log, trace, metric** | **68.44 GB** | **337** | **73** | **28** |
>
> Note the synthetic datasets are generated via simulator algorithms. Thus, the detailed number of the data size, case number, etc. are marked as N/A.
>
> In addition, due to the **proprietary nature of RCA data** (as discussed in **Q1-A1**), researchers often directly use proprietary system datasets for evaluation without open-sourcing them (e.g., CIRCA [1], MULAN [9], CausIL [10], FaultInsight [11], ε-Diagnosis [12]), making it challenging for other researchers to replicate the experiments and conduct comprehensive comparisons with their approaches. For example, while CIRCA additionally includes a database system in its evaluation, the corresponding dataset is inaccessible. To this end, we propose OpenRCA, the largest and most diverse open-source benchmark for RCA. It is also the only goal-driven RCA benchmark with rigorous data quality standards, designed to evaluate LLMs’ RCA capabilities across diverse needs. We believe OpenRCA can largely facilitate the development and evaluation of RCA approaches in our community.
>
> **Reference**
>
> [1] Li, Mingjie, et al. "Causal inference-based root cause analysis for online service systems with intervention recognition." *Proceedings of the 28th ACM SIGKDD Conference on Knowledge Discovery and Data Mining*. 2022.
>
> [2] Liu, Chenghao, et al. "PyRCA: A Library for Metric-based Root Cause Analysis." *arXiv preprint arXiv:2306.11417* (2023).
>
> [3] Ikram, Azam, et al. "Root cause analysis of failures in microservices through causal discovery." *Advances in Neural Information Processing Systems* 35 (2022): 31158-31170.
>
> [4] Wu, Li, et al. "Microrca: Root cause localization of performance issues in microservices." *NOMS 2020-2020 IEEE/IFIP Network Operations and Management Symposium*. IEEE, 2020.
>
> [5] Yu, Guangba, et al. "Microrank: End-to-end latency issue localization with extended spectrum analysis in microservice environments." *Proceedings of the Web Conference 2021*. 2021.
>
> [6] Hou, Chuanjia, et al. "Diagnosing performance issues in microservices with heterogeneous data source." *2021  IEEE Intl Conf on Parallel & Distributed Processing with Applications, Big Data & Cloud Computing, Sustainable Computing  & Communications, Social Computing & Networking  (ISPA/BDCloud/SocialCom/SustainCom)*. IEEE, 2021.
>
> [7] Lee, Cheryl, et al. "Eadro: An end-to-end troubleshooting framework for microservices on multi-source data." *2023 IEEE/ACM 45th International Conference on Software Engineering (ICSE)*. IEEE, 2023.
>
> [8] Yu, Guangba, et al. "Nezha: Interpretable fine-grained root causes analysis for microservices on  multi-modal observability data." *Proceedings of the 31st ACM Joint  European Software Engineering Conference and Symposium on the Foundations of Software Engineering*. 2023.
>
> [9] Zheng, Lecheng, et al. "MULAN: Multi-modal Causal Structure Learning and Root Cause Analysis for Microservice Systems." *Proceedings of the ACM on Web Conference 2024*. 2024.
>
> [10] Chakraborty, Sarthak, et al. "Causil: Causal graph for instance level microservice data." *Proceedings of the ACM Web Conference 2023*. 2023.
>
> [11] Bi, Tingzhu, et al. "FaultInsight: Interpreting Hyperscale Data Center Host Faults." *Proceedings of the 30th ACM SIGKDD Conference on Knowledge Discovery and Data Mining*. 2024.
>
> [12] Shan, Huasong, et al. "?-diagnosis: Unsupervised and real-time diagnosis of small-window long-tail latency in large-scale microservice platforms." *The World Wide Web Conference*. 2019.

---

### Official Review · Reviewer_KxEv · 2024-11-08

**Soundness:** 3
**Presentation:** 3
**Contribution:** 3
**Rating:** 8
**Confidence:** 5

**Summary:**

This paper introduces OpenRCA, a novel benchmark and evaluation framework aimed at assessing the ability of large language models (LLMs) to perform root cause analysis (RCA) in real-world software systems, specifically focusing on the post-development phase, which has been largely overlooked in current research. The benchmark consists of 335 failure cases from three enterprise software systems, accompanied by over 68 GB of telemetry data (logs, metrics, and traces), challenging LLMs to identify the root causes of software failures by reasoning across complex, multi-modal data. The paper also proposes RCA-agent, a specialized tool to help LLMs process large volumes of telemetry data. The evaluation results show that current models, even with RCA-agent, struggle to handle these tasks effectively, with the best-performing model (Claude 3.5) solving only 11.34% of the failure cases. This paper makes a valuable contribution by highlighting the challenges and potential of LLMs in post-deployment software maintenance and lays a foundation for future research in automated root cause analysis.

**Strengths:**

The paper demonstrates several key strengths. First, it offers a clear and detailed evaluation, presenting comprehensive results that highlight how different models perform on the OpenRCA benchmark. The inclusion of various methods, such as RCA-agent and different sampling techniques, alongside a breakdown of performance by system complexity and task difficulty, provides valuable insights. Second, the model comparison between proprietary and open-source LLMs is highly informative, showcasing the limitations of current open-source models like Llama 3.1 compared to more advanced proprietary models like Claude 3.5 and GPT-4o, effectively setting the stage for future improvements. The paper also includes a strong task complexity and system analysis, showing how model performance correlates with system complexity and the number of root cause elements, offering practical insights for future research. The reasoning length and error tolerance analysis provides an interesting perspective on how LLMs' performance can be improved with longer reasoning and better error handling. Finally, the real-world applicability is demonstrated through a detailed case study, illustrating how the OpenRCA benchmark can be used in practice to solve root cause analysis tasks, thus bridging the gap between theory and real-world challenges.

**Weaknesses:**

Insufficient Contextualization of Problem Importance:
Lack of Quantitative Evidence: The paper mentions that software failures cost "billions of dollars," but does not provide specific references or data to substantiate this claim. For instance, Figure 1 cites the financial impact without attributing it to any source.

Actionable Insight: Provide concrete data and citations from industry reports or academic studies that quantify the financial and operational impact of software failures. For example, referencing studies from Gartner or IEEE could strengthen the argument and give readers a clearer understanding of the problem's magnitude.

Inadequate Literature Review on Existing RCA Benchmarks:
Questionable Claim in Section 2.2: The assertion that "RCA is a critical step in the software development lifecycle but often lacks practical benchmarks" may not fully reflect the current state of research. There are existing benchmarks and tools for RCA in software engineering.

Actionable Insight: Expand the literature review to include a comprehensive analysis of existing RCA benchmarks and tools. Discuss works like the MIMIC dataset in healthcare analytics or other RCA frameworks in software engineering to highlight how OpenRCA differentiates itself and fills existing gaps.

Limited Discussion on Broader Impact and Applications:
Underdeveloped Broader Impact Section: The paper does not sufficiently elaborate on how OpenRCA can improve software maintenance, quality assurance, or developer workflows.

Superficial Analysis of Low Model Performance:
Lack of Deep Dive into Results: The observation that "current models can only handle the simplest cases" lacks a thorough analysis. The paper does not explore why models performed poorly—whether it's due to limitations in model architecture, data complexity, or other factors.
Actionable Insight: Conduct a detailed error analysis to identify the root causes of low model performance. Discuss whether issues stem from the models' inability to process long-context data, difficulties in reasoning over heterogeneous data types, or shortcomings in handling complex software dependencies. Providing specific examples where models failed can offer valuable insights.

Overreliance on Synthetic Queries and Lack of Real-World Data:
Synthetic Nature of Failure Reports: The queries are synthesized based on common failure reports, which may not capture the complexity and nuances of real-world scenarios.

Complex Tables and Figures Without Adequate Explanation: Some tables (e.g., Table 2) and figures (e.g., Figure 5) are dense and may be difficult to interpret without additional context. References to Appendix A suggest that crucial information is relegated to the appendices.

Limited Diversity of the Dataset:
Narrow Focus on Distributed Systems: The dataset comprises failures from only three enterprise distributed software systems, potentially limiting the applicability of the benchmark to other system architectures like monolithic or embedded systems.

Absence of Ethical Considerations and Potential Risks:
No Discussion on Ethical Implications: The paper does not address potential ethical concerns, such as the consequences of incorrect root cause identification in critical systems, which could lead to misdiagnosis and exacerbate issues.

**Questions:**

Can you provide more details on the selection process for the failure cases in OpenRCA?

Clarification Needed: The paper mentions 335 failure cases from three enterprise systems but does not describe how these cases were selected. Were they chosen based on frequency, severity, or another criterion?
Suggestion: Please clarify the criteria used to select these cases and explain how representative they are of real-world failure patterns.

What are the specific challenges faced by models when handling multiple root cause elements?

Clarification Needed: The paper observes a significant drop in accuracy as the number of required root cause elements increases. Is this due to limitations in context window size, inability to handle heterogeneous data, or something else?
Suggestion: A more detailed explanation of the challenges could help readers understand the underlying issues and suggest pathways for improvement.

**Details Of Ethics Concerns:**

The ethical concerns surrounding OpenRCA and RCA-agent include the risks of misdiagnosis in critical systems, potential biases from synthetic data, privacy issues with real-world telemetry, and over-reliance on automated RCA without human oversight. Additionally, accountability is a concern due to limited transparency in RCA-agent's reasoning, while model biases and error tolerance constraints may skew fault diagnosis. Environmental impact from resource demands and potential security risks if RCA outputs are accessed by malicious actors also raise ethical questions. Addressing these issues by enhancing transparency, ensuring data privacy, promoting human-in-the-loop approaches, and minimizing resource usage will foster responsible and ethical use of RCA-agent in real-world applications.

---

> ### Author Response · Authors · 2024-11-21
>
> Thank you so much for your comprehensive review. Please refer to our clarifications below:
>
> **Q1**: *The paper mentions that software failures cost "billions of dollars",  but does not provide specific references or data to substantiate this claim.*
>
> **A1**: We have provided two recent pieces of evidence in 2024 to support this claim:
>
> 1. Incidents involving Azure and CrowdStrike caused global Windows blue screen errors due to software service updates, resulting in an estimated cost of at least $10 billion [1,2].
> 2. Misconfigurations of cloud service systems in Google Cloud and UniSuper led to the unintended deletion of accounts valued at $135 billion [3,4].
>
> These reference reports have been incorporated into Sec. 1 in the revised version.
>
> **Q2**: *The assertion that "RCA is a critical step in the software development lifecycle but often lacks practical benchmarks" may not fully reflect the current state of research.*
>
> **A2**: We would like to clarify that current RCA datasets are either synthetic or small-scale, typically containing only dozens of failure cases from a single system [5,6]. This limits their effectiveness for comprehensive large-scale evaluations. OpenRCA provides hundreds of failures collected from three real-world software systems. This paves the way for solving more practical RCA problems at scale.
>
> Furthermore, as discussed in the "Goal-driven Task Design" section of Sec. 2.2, we introduced several existing evaluation datasets and highlighted the key differences between OpenRCA and these datasets. A major limitation of current datasets is their focus on a single goal (e.g., identifying only the originating component), which encourages RCA methods to be narrowly tailored to each dataset, compromising generalizability. Since our goal is to evaluate *whether LLMs can handle RCA*, this limitation threatens the validity of benchmarks in more general settings.
>
> We have revised Sec. 2.2 to clarify these distinctions and avoid potential misunderstandings. Additionally, regarding the reviewer’s suggestion to discuss the MIMIC dataset, we would like to clarify that MIMIC is a medical imaging benchmark, not a root cause analysis dataset. Thus, we did not include it in our revision.
>
> **Q3**: *The paper does not sufficiently elaborate on how OpenRCA can improve software maintenance, quality assurance, or developer workflows.*
>
> **A3**: As discussed in Sec. 1, LLM-based methods, such as software development assistants (e.g., Copilot, Cursor), have transformed the paradigm of coding. Therefore, we believe that leveraging LLMs to address complex post-development challenges is also a promising direction. The goal of OpenRCA is to set up a comprehensive benchmark and encourage the development of various automatic RCA methods or assistants (e.g., using LLMs or Agents) to facilitate the developer’s productivity and reduce the costs caused by service downtime—much like how coding assistants have significantly supported developers during programming.
>
> **Q4**: *The observation that "current models can only handle the simplest cases" lacks a thorough analysis. The paper does not explore why models performed poorly—whether it's due to limitations in model architecture, data complexity, or other factors.*
>
> **A4**: Since the prompting-based LLM baseline directly generates a final answer to the query without intermediate outputs, it is challenging to analyze the reasoning process of these sampling-based methods. Therefore, as demonstrated in Appendix C1 and C3, we conducted a case study using Chain-of-Thought prompting to explicitly output the model’s reasoning process. We observed that the model often struggles with maintaining intermediate conclusions from earlier steps in a lengthy reasoning chain. This frequently results in a final conclusion that directly contradicts an earlier intermediate conclusion.  We consider this issue arises from the model's current limitations in handling excessively long inputs encoded in non-natural language formats, which hinders its ability to effectively focus on the reasoning context in subsequent steps. We also investigate the limitation of LLMs when working as an agent beyond the vanilla model. More details can be found in Sec. 5.

---

> > ### Comment · Reviewer_KxEv · 2024-11-21
> >
> > Agreed upon explanations and revisions.

---

> ### Author Response · Authors · 2024-11-21
>
> **Q5**: *The queries are synthesized based on common failure reports, which may not capture the complexity and nuances of real-world scenarios.*
>
> **A5**: As mentioned in Sec. 2.4 and Sec. 8, due to privacy concerns, real-world incident reports or queries from companies are often unavailable. Therefore, we opted to design more general queries based on common domain knowledge in RCA, i.e., synthesizing queries based on the root cause time, component, and reason. It is important to note that these three elements are the most common focus areas in real-world RCA issues. In traditional RCA tasks, different approaches often focus on different elements [5,6], with some only considering one of the three and others possibly addressing multiple elements. Given that the design of these three elements directly influences the objectives of RCA tasks, we chose to use them to synthesize queries. Moreover, this synthetic strategy offers an additional advantage: it allows for high customization of queries if more details of the specific systems can be provided by the users, and can be scaled up to incorporate new datasets.
>
> **Q6**: *Some tables (e.g., Table 2) and figures (e.g., Figure 5) are dense and may be difficult to interpret without additional context. References to Appendix A suggest that crucial information is relegated to the appendices.*
>
> **A6**: Thank you for the suggestion. Due to space constraints, we initially moved these explanations to Appendix A. We have now moved some information regarding system differences from Appendix A to Sec. 5, where Table 2 and Figure 5 are discussed, to provide more valuable context. We also added more explanations to the Table 2 and Figure 5 in Sec.5 text. Please refer to the revised manuscript for details.
>
> **Q7**: *The dataset comprises failures from only three enterprise distributed software systems, potentially limiting the applicability of the benchmark to other system architectures like monolithic or embedded systems.*
>
> **A7**: The proprietary nature of telemetry and system failure data makes real-world datasets rare. Companies are often unwilling to share private data or invest in anonymization without clear incentives. Given that mainstream service systems today are typically distributed, as monolithic systems are outdated in many scenarios, we were only able to collect three datasets from distributed systems. Additionally, embedded systems often differ from traditional software systems, focusing more on lower-level hardware than software, and may thus fall outside the scope of OpenRCA.
>
> Nevertheless, we still believe these three distributed systems are diverse representatives, as the distributed systems also have various detailed architectures: the Telecom system uses a distributed cloud database architecture, and both Bank and Market use microservice architectures. In addition, they also represent three systems in much different scenarios. To further enhance the applicability of OpenRCA, we will also keep updating OpenRCA to include more systems as discussed in Sec.8.
>
> **Q8**: *The ethical concerns surrounding OpenRCA and RCA-agent include the risks of misdiagnosis in critical systems, potential biases from synthetic data, privacy issues with real-world telemetry, and over-reliance on automated RCA without human oversight. Additionally, accountability is a concern due to limited transparency in RCA-agent's reasoning, while model biases and error tolerance constraints may skew fault diagnosis. Environmental impact from resource demands and potential security risks if RCA outputs are accessed by malicious actors also raise ethical questions. Addressing these issues by enhancing transparency, ensuring data privacy, promoting human-in-the-loop approaches, and minimizing resource usage will foster responsible and ethical use of RCA-agent in real-world applications.*
>
> **A8**: We would like to clarify the following points:
>
> 1. **Bias/Privacy concerns of OpenRCA:** OpenRCA is an open-sourced benchmark designed to evaluate and compare RCA techniques. The telemetry data used are anonymized, labeled, and open-sourced by their respective companies, mitigating concerns related to privacy or bias. While the queries are synthesized, as discussed in Sec. 2.4, they are generated based on patterns observed in real-world failure queries, minimizing potential bias compared to actual scenarios.
> 2. **Misdiagnosis/Transparency concerns of RCA-Agent:** RCA-Agent serves solely as a baseline for the OpenRCA benchmark and is not intended for deployment in real systems, ensuring it poses no risk to critical operations. Finally, both the benchmark data and the agent's source code are fully open-sourced, ensuring transparency. Thus, we believe that the misdiagnosis and transparency concerns could be mitigated.

---

> > ### Comment · Reviewer_KxEv · 2024-11-21
> >
> > Agreed upon explanations and revisions.

---

> > > ### Author Response · Authors · 2024-11-22
> > > **Thank you!**
> > >
> > > Thank you for your prompt reply. We would be happy to provide more clarifications if needed. If you find our responses satisfactory, we kindly request the reviewer to consider raising the score. Thank you.

---

> > > > ### Comment · Reviewer_KxEv · 2024-11-27
> > > >
> > > > Yes, I have updated your score positively.

---

> ### Author Response · Authors · 2024-11-21
>
> **Q9**: *Can you provide more details on the selection process for the failure cases in OpenRCA? The paper mentions 335 failure cases from three enterprise systems but does not describe how these cases were selected. Were they chosen based on frequency, severity, or another criterion?*
>
> **A9**: As detailed in Sec. 2.4, our selection process for the 335 failure cases in OpenRCA was guided by considerations for *system diversity*, *data balance*, and *data quality* (i.e., whether the labels align with the telemetry data). We began by ensuring diversity through the selection of three distinct enterprise systems in various application scenarios: a telecom database, a banking system, and an online market system. To achieve data balance, we downsampled the datasets to ensure comparable scales across systems, avoiding potential biases. We also focused on data quality by calibrating the dataset to confirm that telemetry data matched the root cause labels, involving expert verification. This rigorous filtering ensured that only the most reliable and representative failure cases were included.
>
> **Q10**: *What are the specific challenges faced by models when handling multiple root cause elements? The paper observes a significant drop in accuracy as the number of required root cause elements increases. Is this due to limitations in context window size, inability to handle heterogeneous data, or something else?*
>
> **A10**: Identifying multiple elements in system failures requires more comprehensive and precise analysis compared to identifying a single element, making it a finer-grained RCA task. For instance, in a simplified scenario where a single component exhibits clear anomalies (e.g., increased CPU or memory usage), a model can easily identify the root cause component. However, if the task is to identify both the component and the reason, the model must further analyze the underlying cause. This additional complexity places higher demands on the model's reasoning capabilities, making the task significantly more challenging.
>
> **Reference**
>
> [1] https://en.wikipedia.org/wiki/2024_CrowdStrike-related_IT_outages
>
> [2] https://blogs.microsoft.com/blog/2024/07/20/helping-our-customers-through-the-crowdstrike-outage/
>
> [3] https://www.unisuper.com.au/about-us/media-centre/2024/a-joint-statement-from-unisuper-and-google-cloud
>
> [4] https://www.arcserve.com/blog/human-error-and-google-cloud-135-billion-account-deletion-how-avoid-costly-downtime-and
>
> [5] Li, Mingjie, et al. "Causal inference-based root cause analysis for online service systems with intervention recognition." Proceedings of the 28th ACM SIGKDD Conference on Knowledge Discovery and Data Mining. 2022.
>
> [6] Ikram, Azam, et al. "Root cause analysis of failures in microservices through causal discovery." Advances in Neural Information Processing Systems 35 (2022): 31158-31170.

---

> > ### Comment · Reviewer_KxEv · 2024-11-21
> >
> > Agreed upon explanations and revisions.

---

> ### Author Response · Authors · 2024-11-27
> **Thank you!**
>
> We are glad that our response clarified your questions. Thank you again for your comprehensive review!

---

### Author Response · Authors · 2024-12-04
**Summary of Rebuttal**

We would like to thank all the reviewers for their detailed review. We summarize the key clarifications&modifications and additional experiments&analysis as follows:

### Clarifications&Modifications

1. **[Reviewer RqHx]** Clarified the insight of OpenRCA’s baseline choice. We also provided a [general response](https://openreview.net/forum?id=M4qNIzQYpd&noteId=KxYY0dx1Wy) to corresponding questions.
2. **[Reviewer KxEv, RSze]** Highlighted the size, scalability, diversity, and the nature of complex data input of OpenRCA.
3. **[Reviewer KxEv, RSze]** Explained the specific challenges of handling multiple elements and the reason for using element number to categorize task complexity.
4. **[Reviewer KxEv]** Explained the details of OpenRCA, including its real-world impact, performance, and criteria of data selection and query construction.
5.  **[Reviewer KxEv]** Explained the potential ethics concerns of OpenRCA.
6. **[Reviewer 41p3]** Explained the reason why OpenRCA does not require the LLMs to identify whether the root cause is identifiable.
7. **[Reviewer KxEv]** Added extra papers and reports as references in Sec. 1 and Sec. 2.
8. **[Reviewer KxEv, 41p3]** Polished the discussions and captions to the tables and figures in the manuscript.
9. **[Reviewer RSze, RqHx]** Corrected the typo in the manuscript.

### Experiments&Analysis

1. **[Reviewer 41p3]** Provided the raw results and discussions of GPT/Claude’s performance on both oracle and agent settings.
2. **[Reviewer 41p3]** Conducted the experimental analysis of majority voting on both oracle and agent settings.
3. **[Reviewer 41p3]** Conducted the experimental analysis of individually localizing each root cause element across all tasks.
4. **[Reviewer RSze]** Provided the raw results and discussions of agent’s performance with different reasoning steps.
5. **[Reviewer RSze]** Provided the statistical data to illustrate the size and diversity of OpenRCA.
6. **[Reviewer RqHx]** Conducted the comparison between agents and vanilla LLMs with the exactly same prompts.

---

### Meta-Review · Area_Chair_e445 · 2024-12-19

**Metareview:**

This paper proposes a benchmark dataset, OpenRCA, for evaluating the ability of LLMs to identify the root cause of software failures. Various LLMs are evaluated and the results show current models can only handle the simplest cases. The proposed benchmark dataset is interesting and it may be a valuable resource for future research. The experiments are comprehensive. The reviewers raised a few concerns about presentation and evaluation, and the authors addressed most of the concerns during the rebuttal period.

**Additional Comments On Reviewer Discussion:**

Most reviewers increased their scores during the rebuttal period.

---

### Decision · Program_Chairs · 2025-01-22

Accept (Poster)